# Boosting electrochemical oxygen reduction to hydrogen peroxide coupled with organic oxidation

Yining Sun[1], Kui Fan[1], Jinze Li[1], Lei Wang[1], Yusen Yang [1,2], Zhenhua Li [1,2] ✉, Mingfei Shao [1,2] ✉ & Xue Duan[1,2]

The electrochemical oxygen reduction reaction (ORR) to produce hydrogen peroxide ($H_2O_2$) is appealing due to its sustainability. However, its efficiency is compromised by the competing 4e$^-$ ORR pathway. In this work, we report a hierarchical carbon nanosheet array electrode with a single-atom Ni catalyst synthesized using organic molecule-intercalated layered double hydroxides as precursors. The electrode exhibits excellent 2e$^-$ ORR performance under alkaline conditions and achieves $H_2O_2$ yield rates of 0.73 mol $g_{cat}^{-1}$ h$^{-1}$ in the H-cell and 5.48 mol $g_{cat}^{-1}$ h$^{-1}$ in the flow cell, outperforming most reported catalysts. The experimental results show that the Ni atoms selectively adsorb $O_2$, while carbon nanosheets generate reactive hydrogen species, synergistically enhancing $H_2O_2$ production. Furthermore, a coupling reaction system integrating the 2e$^-$ ORR with ethylene glycol oxidation significantly enhances $H_2O_2$ yield rate to 7.30 mol $g_{cat}^{-1}$ h$^{-1}$ while producing valuable glycolic acid. Moreover, we convert alkaline electrolyte containing $H_2O_2$ directly into the downstream product sodium perborate to reduce the separation cost further. Techno-economic analysis validates the economic viability of this system.

Hydrogen peroxide ($H_2O_2$) is an important and eco-friendly chemical that is widely used in sewage treatment and paper bleaching, and also serves as a green oxidant to facilitate the synthesis of organic oxygenated chemicals such as caprolactam and propylene oxide[1–7]. Consequently, the global capacity requirements for $H_2O_2$ have progressively escalated in recent years. Currently, the energy-intensive anthraquinone cycling method is the mainstream strategy for large-scale production of $H_2O_2$ in industry. This method utilizes palladium as the catalyst and yields about 70 wt% $H_2O_2$ through hydrogenation/oxidation of anthraquinone[8,9]. However, the generation of organic waste streams during the synthesis process and the high cost of transporting the $H_2O_2$ product are unavoidable drawbacks of this method[10]. Directly mixing hydrogen and oxygen at a certain pressure ( > 2.0 MPa) represents a more straightforward method for synthesizing $H_2O_2$. However, this approach exhibits less selective (<30%) and raises safety concerns[11–14]. A more benign alternative is urgently needed.

The electrochemical oxygen reduction reaction (ORR) via a 2e$^-$ pathway (alkaline condition 2e$^-$ ORR: $O_2 + H_2O + 2e^- \rightarrow HO_2^- + OH^-$) has been recognized as a promising alternative method for $H_2O_2$ production due to its environmental friendliness, safety, and sustainability[15]. Nevertheless, this process still suffers from low Faradaic efficiency (FE) and a low $H_2O_2$ yield rate due to the competitive 4e$^-$ ORR process (alkaline condition 4e$^-$ ORR: $O_2 + 2H_2O + 4e^- \rightarrow 4OH^-$)[16,17]. A typical 2e$^-$ ORR reaction under alkaline conditions consists of (i) $O_2 + * \rightarrow *O_2$ ($O_2$ adsorption on the reactive site; the end-on configuration is favourable for $H_2O_2$ production), (ii) $*O_2 + H_2O + e^- \rightarrow *OOH + OH^-$ (including water splitting to produce reactive hydrogen species (H*) and the subsequent hydrogenation step), and (iii) $*OOH + e^- \rightarrow HO_2^- + *$ (the desorption step from the catalyst surface). Among them, H* generation and $O_2$ selective adsorption play important roles in the 2e$^-$ ORR. However, catalysts containing a single reactive site face challenges due to the correlation and competition among multiple reaction steps on the

[1]State Key Laboratory of Chemical Resource Engineering, Beijing University of Chemical Technology, Beijing 100029, China. [2]Quzhou Institute for Innovation in Resource Chemical Engineering, Quzhou 324000, China. ✉e-mail: LZH0307@mail.buct.edu.cn; shaomf@mail.buct.edu.cn

same reactive sites. Optimizing one step may result in suboptimal conditions for other steps. Therefore, efficient electrocatalysts need to be designed to synergistically regulate H* generation and $O_2$ adsorption capacity to promote the $2e^-$ ORR to produce $H_2O_2$.

Single-atom catalysts (SACs) have emerged as a hotspot in the ORR field, promoting the adsorption of $O_2$ through an end-on configuration and facilitating a suitable *OOH binding energy formation[18–24]. However, most laboratory-scale SACs for $O_2$ to $H_2O_2$ employ powder carriers with limited electrochemical properties[25,26]. In addition, the power consumption required to operate the system and its economic viability cannot be disregarded. Substituting the conventional oxygen evolution reaction (OER) with the thermodynamically more favourable organic oxidation and electrically pairing the $2e^-$ ORR can effectively mitigate these problems[27–29]. Herein, we fabricated a hierarchical carbon nanosheet array electrode containing a single-atom Ni catalyst (Ni-SAC) by using organic molecule-intercalated layered double hydroxides (LDHs) as precursors. The electrode exhibits outstanding $2e^-$ ORR performance under alkaline conditions. For example, the Ni-SAC electrode exhibits a high FE (up to 89.02%) and attains a $H_2O_2$ yield rate of up to 0.73 mol $g_{cat}^{-1}$ $h^{-1}$, overwhelming most related works. We further applied a Ni-SAC electrode in a two-electrode flow cell, achieving an industrial current density of $-261.73$ mA $cm^{-2}$ at a cell voltage of $-2$ V, with an FE of up to 91.36% and a $H_2O_2$ yield rate of up to 5.48 mol $g_{cat}^{-1}$ $h^{-1}$. The underlying mechanism for $H_2O_2$ production over Ni-SAC was verified by electron spin resonance (ESR), kinetic isotope effect (KIE), $O_2$ temperature-programmed desorption ($O_2$-TPD), and in situ Fourier transform infrared spectroscopy (FTIR). The results indicate that the single Ni atom in Ni-SAC acts as a site for the selective adsorption of $O_2$, while the carbon nanosheets (CNS) generate reactive hydrogen species (H*) during $O_2$ reduction. This cooperative mechanism significantly boosts the high-throughput production of $H_2O_2$. Finally, a coupling reaction system was constructed by pairing the $2e^-$ ORR at the cathode with the ethylene glycol oxidation reaction at the anode, achieving the simultaneous production of $H_2O_2$ (FE$_{max}$= 99.83%) and high-value-added glycolic acid (FE$_{max}$= 97.80%). We also attempted to convert the alkaline electrolyte containing $H_2O_2$ directly into the downstream sodium perborate (SPB) product to further reduce the separation cost. The techno-economic evaluation shows that the coupling system has a higher profit margin of $15.65*10^6$ \$/year, enabling broad application prospects.

## Results

### Characterization of the single-atom Ni catalyst

An atomic Ni-modified carbon nanosheet array (Ni-SAC) electrode was synthesized via a confinement strategy[30]. As shown in Fig. 1a and Supplementary Fig. 1, a NiAl-LDH array with metanilic acid intercalation (denoted as NiAl-LDH(MA)) was vertically grown on hydrophilic carbon cloth via a hydrothermal method. Then, a pyrolysis process was performed to obtain the Ni nanoparticles and single-atom embedded carbon nanosheets (denoted as Ni-CNS), with the intercalated MA converted to carbon (Supplementary Fig. 2). This process was followed by further acid etching to obtain Ni-SAC. Carbon nanosheets (CNS) electrode with almost no Ni was also obtained by acid etching the Ni-CNS in 1 M HCl under heating conditions (60 °C) for 12 h (Supplementary Fig. 3 and Supplementary Table 1). CNS and Ni-CNS were used as control samples of Ni-SAC to verify the critical role of the single-atom Ni in the $2e^-$ ORR. Figure 1b shows the X-ray diffraction (XRD) patterns of the as-prepared samples. The XRD pattern of NiAl-LDH(MA) shows a typical diffraction (003) peak at 5.82° (yellow curve in Fig. 1b), indicating an interlayer spacing of 1.52 nm, which correlates well with the MA-intercalated LDH[31,32]. For the Ni-CNS sample, the original characteristic peaks of LDH disappear after the pyrolysis process (blue curve in Fig. 1b) and are replaced by the peaks of metallic Ni (PDF#04-0850) at approximately 44.5° (corresponding to the (111)

crystal planes) and 51.9° (corresponding to the (200) crystal planes), as well as the characteristic peak of graphitic carbon (002) at approximately 25°[33]. After acid etching treatment, the characteristic peaks of the metallic Ni disappear for the Ni-SAC sample, leaving the (002) and (100) peaks of graphitic carbon at approximately 25° and 44°, respectively (red curve in Fig. 1b)[19,34,35]. The scanning electron microscopy (SEM) image of the Ni-SAC sample shows a typical porous nanosheet array structure vertically grown on carbon cloth. The nanopores can be clearly observed on the carbon nanosheets and are formed by acid etching of the Ni nanoparticles (Fig. 1c and Supplementary Fig. 4). High-resolution transmission electron microscopy (HRTEM) further verifies the porous nanosheet structure of Ni-SAC (Fig. 1d and Supplementary Fig. 5). Moreover, no visible Ni nanoparticles are observed in the carbon nanosheets. The above results indicate that the acid etching process can effectively remove the aggregated and unstable Ni nanoparticles. The HRTEM images of the Ni-CNS and CNS are shown in Supplementary Figs. 6 and 7. The atomic-resolution high-angle annular dark-field scanning transmission electron microscopy (HAADF-STEM) images show that scattered bright spots are uniformly dispersed on the carbon nanosheets of the Ni-SAC sample (Fig. 1e and Supplementary Fig. 8). The energy-dispersive X-ray (EDX) mapping results also show the even distribution of Ni in the Ni-SAC catalyst (Supplementary Fig. 9). Notably, this strategy enables the easy synthesis of large-area electrocatalysts (Supplementary Fig. 10), demonstrating potential for further scale-up preparation of Ni-SAC.

We then performed X-ray photoelectron spectroscopy (XPS) to elucidate the chemical composition and structure of Ni-SAC. As shown in Supplementary Fig. 11, Ni, Al, C, N, O and S were detected in the full XPS spectrum. Note that the proportion of Ni in the Ni-SAC sample is markedly lower than that in the Ni-CNS sample due to the acid etching of the Ni nanoparticles. This finding is consistent with the inductively coupled plasma optical emission spectroscopy (ICP-OES) results (Supplementary Table 2). The characteristic peak at 853.1 eV in the Ni $2p_{3/2}$ spectrum of the Ni-CNS sample is assigned to zero-valent nickel (Supplementary Fig. 12)[36]. While the binding energy of Ni-SAC shows a positive shift of 0.8 eV (to 853.9 eV) relative to Ni-CNS, indicating that Ni in Ni-SAC is in an oxidation state (Fig. 1f). The deconvolution of the N $1s$ spectra for Ni-SAC and Ni-CNS reveal pyridinic–N, Ni–N, pyrrolic–N, and graphitic–N (Supplementary Fig. 13)[35,37]. The peaks located at 161.7 eV, 163.0 eV, 164.8 eV and 168.6 eV in the S $2p$ spectra of Ni-SAC are attributed to Ni–S, C–S–C $2p_{3/2}$, C–S–C $2p_{1/2}$ and C–SO$_x$–C, respectively (Supplementary Fig. 14)[38]. The above XPS results indicate the successful incorporation of N and S into the coordination environment, where single-atom Ni serves as the metal sites. Comprehensive information regarding the parameters and outcomes of XPS fitting can be found in Supplementary Table 3. Subsequently, we employed X-ray absorption near-edge structure (XANES) and extended X-ray absorption fine structure (EXAFS) analyses to investigate the coordination environment of the nickel atoms in Ni-SAC. The Ni K-edge reveals that the position of the white line peak for Ni-SAC is situated in a higher energy region than both the Ni-CNS and Ni foil, which is in accordance with the XPS results demonstrating that the Ni atoms in Ni-SAC are in the oxidation state (Fig. 1g). Fourier transform EXAFS (FT-EXAFS) shows the presence of a scattering path, which is attributed to the Ni–Ni bond (2.20 Å) in the Ni-CNS. However, only a strong peak at 1.75 Å for Ni-SAC is observed (Fig. 1h). Notably, unlike the widely reported Ni–N$_4$ (1.49 Å)[39], the main peak of Ni-SAC is located in a higher $R$-space. We speculate that Ni-SAC is a structure in which Ni is co-coordinated with light heteroatoms (N/S) based on the XPS analysis. The wavelet-transform EXAFS (WT-EXAFS) results show that the maximum intensity of the WT contour plot for Ni-SAC occurs around 4.35 Å$^{-1}$. This value is distinctly different from that of Ni foil (7.5 Å$^{-1}$) and NiO (6.8 Å$^{-1}$), confirming the atomic dispersion of Ni in Ni-SAC (Supplementary Fig. 15). Moreover, the WT contour maximum intensity of the Ni-CNS is located at 7.3 Å$^{-1}$, akin to that of the Ni foil, echoing the presence of zero-valent

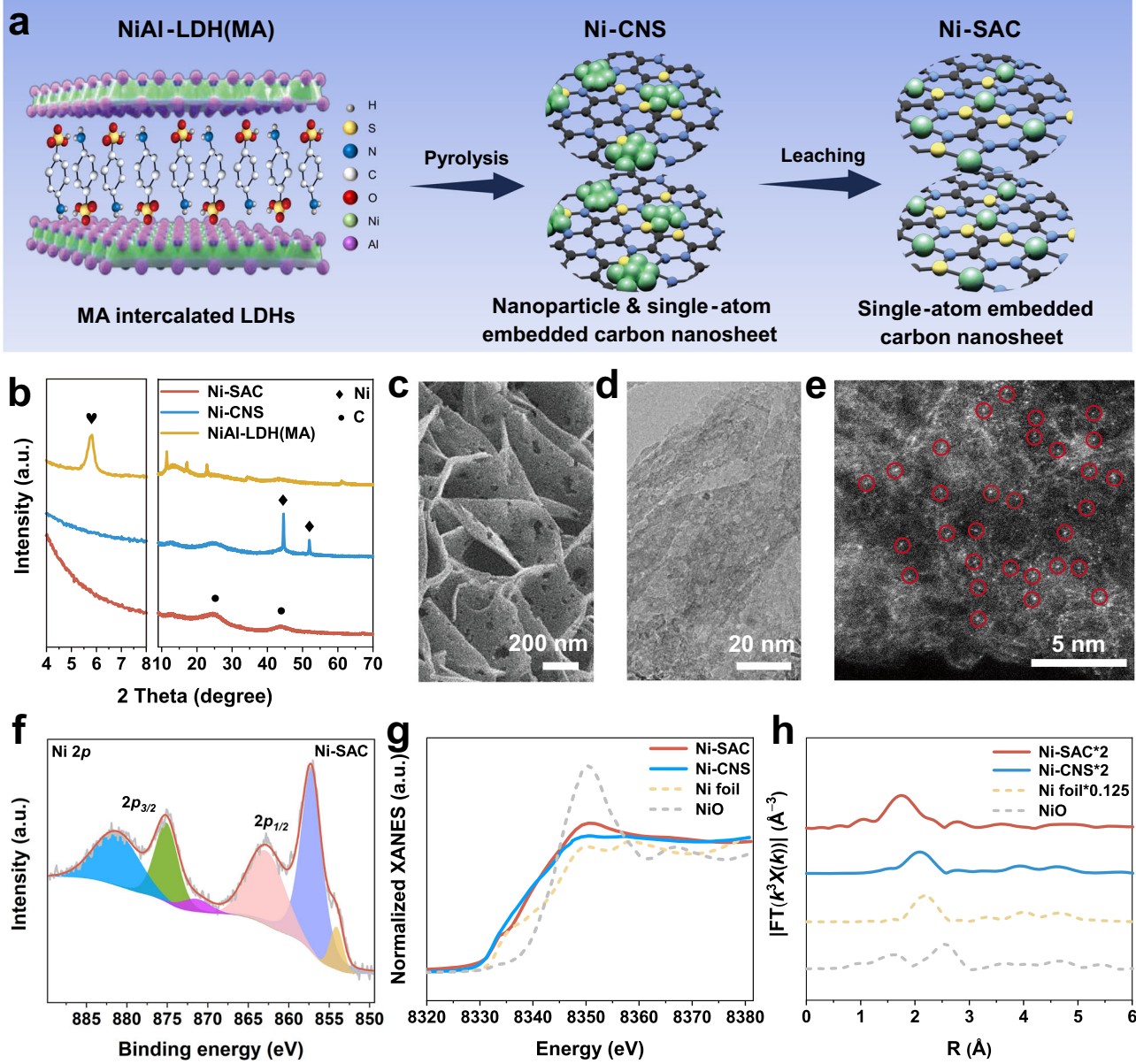

**Fig. 1 | Morphology and structural characterization of single-atom Ni.**
**a** Schematic diagram of the synthesis of Ni-SAC by the confinement synthesis strategy. **b** XRD patterns of NiAl-LDH(MA), Ni-CNS, Ni-SAC. **c** SEM, **d** HRTEM, and **e** HAADF-STEM images of Ni-SAC sample. **f** High-resolution XPS spectra of Ni 2*p* in Ni-SAC. **g** Ni K-edge XANES and the corresponding **h** Fourier transform EXAFS spectra of Ni-CNS and Ni-SAC.

Ni in the XPS analysis. The optimal fitting results of quantitative least-squares EXAFS curve fitting indicate that the Ni atom in Ni-SAC is coordinated with approximately 3.9 heteroatoms (N/S) in the first shell (Supplementary Fig. 16 and Supplementary Table 4).

**Electrochemical ORR to produce $H_2O_2$ over the Ni-SAC electrode**
We then evaluated the 2e⁻ ORR performance of the Ni-SAC electrode to produce $H_2O_2$ using the Ni-CNS as a reference sample. The electrochemical tests were performed in a three-electrode configuration within a divided H-type cell using $O_2$- or $N_2$-saturated 0.1 M KOH solution as the electrolyte (Supplementary Fig. 17). Linear sweep voltammetry (LSV) curves show that the Ni-SAC and Ni-CNS samples exhibit negligible current responses in $N_2$-saturated electrolyte (dash line in Fig. 2a). However, under $O_2$-saturated conditions, the current densities of both the Ni-SAC and Ni-CNS electrodes are clearly enhanced, demonstrating the occurrence of the ORR (solid line in Fig. 2a). Specifically, the Ni-SAC electrode exhibits an onset potential ($E_{onset}$) of 0.857 V vs. RHE (defined

as the potential at a current density of 0.1 mA cm⁻²) and the maximum current density of 37.40 mA cm⁻², both superior to those of the Ni-CNS sample (0.828 V and 16.66 mA cm⁻²). The above results indicate that the Ni-SAC sample has better ORR performance. Note that the $E_{onset}$ of Ni-SAC and Ni-CNS are slightly greater than the thermodynamic theoretical value required for the occurrence of the 2e⁻ ORR under alkaline conditions ($O_2 + H_2O + 2e^- \rightarrow HO_2^- + OH^-$, 0.75 V vs. RHE). The reason for this is potentially the shift in the Nernst potential caused by the low concentration of $H_2O_2$ in the electrolyte, which also may be related to its pH[40]. Moreover, Ni-SAC has a lower Tafel slope than Ni-CNS (79.5 mV dec⁻¹ vs. 146.8 mV dec⁻¹), revealing that Ni-SAC has better reaction kinetics (Supplementary Fig. 18). Compared to Ni-CNS, Ni-SAC has a larger electrochemical active surface area (ECSA), as evidenced by its higher double-layer capacitance ($C_{dl}$) (33.5 mF cm⁻² vs. 21.8 mF cm⁻²). After normalizing the ECSA to the LSV curves, Ni-SAC still exhibits a significant current density, indicating high intrinsic activity (Supplementary Fig. 19).

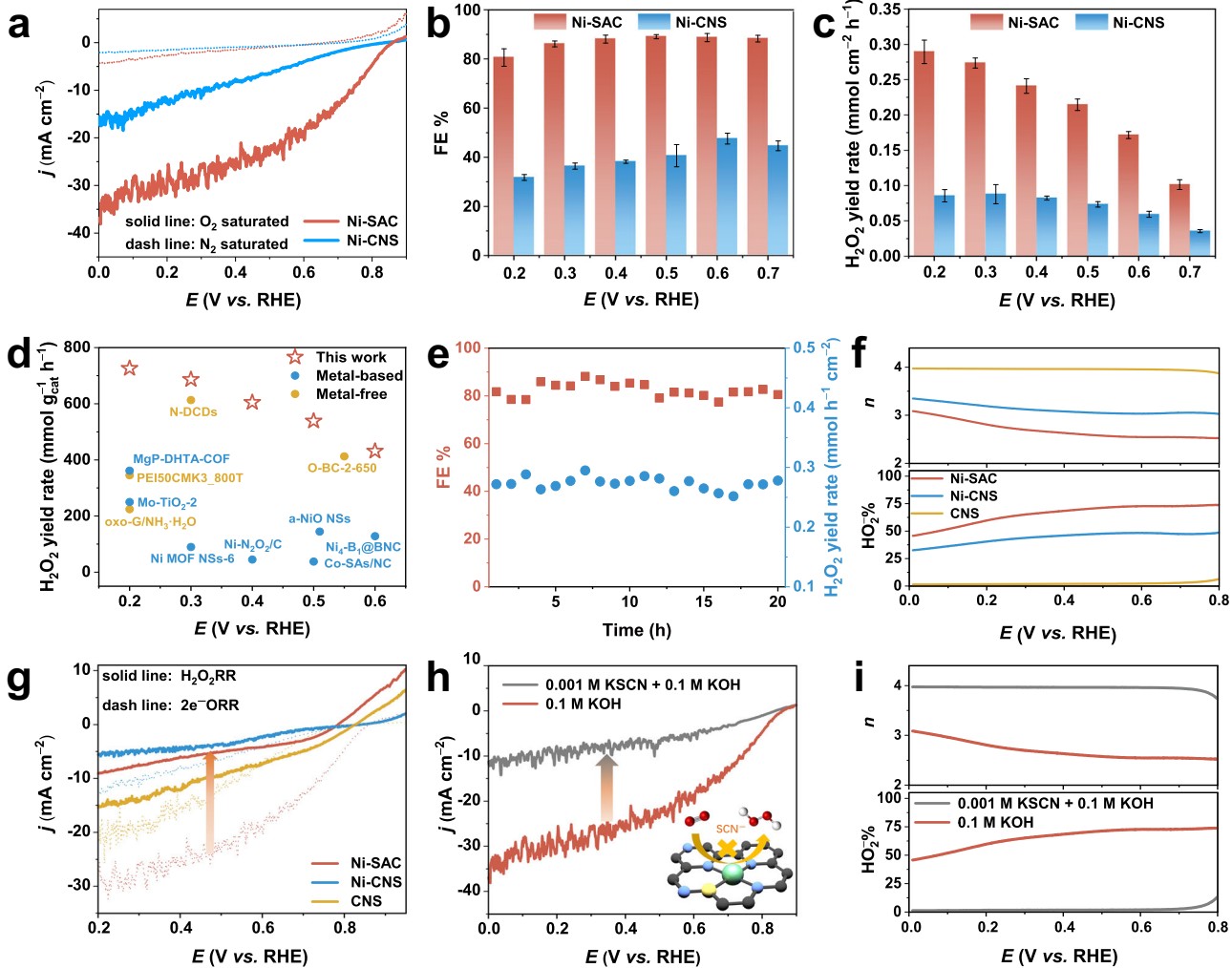

**Fig. 2 | O₂ electroreduction performance of Ni-SAC for the 2e⁻ ORR to produce H₂O₂. a** LSV curves of Ni-SAC and Ni-CNS in 0.1 M O₂/N₂-saturated KOH. **b** Faradaic efficiency (FE) and **c** H₂O₂ yield rate of each sample in the voltage range of 0.2 V to 0.7 V. **d** Comparison of H₂O₂ yield rate (mmol $g_{cat}^{-1}$ h⁻¹) with other reported literature under alkaline condition (0.1 M KOH) in H-cell. **e** Stability test of Ni-SAC at 0.3 V for 20 h. **f** Electron transfer number (*n*) and HO₂⁻ % of Ni-SAC, Ni-CNS, CNS. **g** LSV curves corresponding to H₂O₂RR in N₂-saturated 0.1 M KOH with 10 mM

H₂O₂ solution. **h** Corresponding ORR polarization curves before and after 1 mM KSCN poisoning in 0.1 M O₂-saturated KOH for Ni-SAC. **i** Electron transfer number (*n*) and HO₂⁻ % after poisoning for Ni-SAC. The three-electrode setup performed in the H-cell, RRDE, and RDE have no *iR*-compensation. The error bars are defined as standard deviation, and the centre of each error bar represents the mean value of the corresponding three independent experiments.

We then quantified the accumulated H₂O₂ in the cathode chamber during the ORR process using an ultraviolet-visible (UV-Vis) spectro-photometer to investigate the Faradaic efficiency (FE) and yield rate of H₂O₂. The standard curve for calculating the H₂O₂ concentration is displayed in Supplementary Fig. 20. Figure 2b demonstrates that the FE of the 2e⁻ ORR over the Ni-SAC electrode achieves a maximum of 89.02 ± 0.84% and consistently remains an approximate value of 85% in a wide potential window from 0.3 V to 0.7 V. In contrast, the FE of the Ni-CNS electrode is substantially lower, with a maximum value not exceeding 45%. The yield rate of H₂O₂ over Ni-SAC progressively increases with increasing overpotential, reaching a maximum value of 0.29 ± 0.02 mmol h⁻¹ cm⁻² at 0.2 V vs. RHE, approximately 3.4-fold greater than that of Ni-CNS (Fig. 2c). The developed Ni-SAC electrode also exhibits an evident advantage in the yield rate of H₂O₂ (up to 0.73 mol $g_{cat}^{-1}$ h⁻¹) compared to the currently reported work under alkaline conditions, as shown in Fig. 2d, Supplementary Fig. 21 and Supplementary Table 5. We further tested the long-term stability of Ni-SAC. As shown in Fig. 2e, the Ni-SAC electrode can operate stably for 20 h at 0.3 V vs. RHE. Results from SEM, HAADF-STEM, and XRD results after the stability test show that the structural integrity of the

nanoarrays and the dispersion of the single atoms are maintained (Supplementary Figs. 22–24).

To rationalize the role of single-atom Ni in enhancing the selectivity of the 2e⁻ ORR, we employed rotating ring-disk electrode (RRDE) tests in O₂-saturated 0.1 M KOH to access the 2e⁻ ORR selectivity over different catalysts during the ORR. Owing to the unsuitability of integrated electrodes for RRDE studies, we scraped powder samples from these electrodes to gather evidence regarding the intrinsic activity of the catalyst. During the RRDE test, the H₂O₂ generated at the disk electrode diffuses to the ring electrode at a rotation speed of 1600 rpm and is oxidized at a ring voltage of 1.2 V vs. RHE. As shown in Fig. 2f and Supplementary Fig. 25, the Ni-SAC sample exhibits a low electron transfer number (*n*) value close to 2.5 in the potential window from 0.2 to 0.7 V vs. RHE with HO₂⁻ selectivity (%) up to 73%. Note that the HO₂⁻ % is lower than that in H-cell tests due to the limited mass transfer of the catalyst in the powder state relative to the integrated electrode. The RDE measurements align well with those of the RRDE, indicating high selectivity for the 2e⁻ ORR over Ni-SAC (Supplementary Fig. 26). The *n* value of Ni-CNS is evidently greater than that of Ni-SAC (e.g., 3 vs. 2.5 at 0.8 V vs. RHE), emphasizing the important role of single Ni atoms

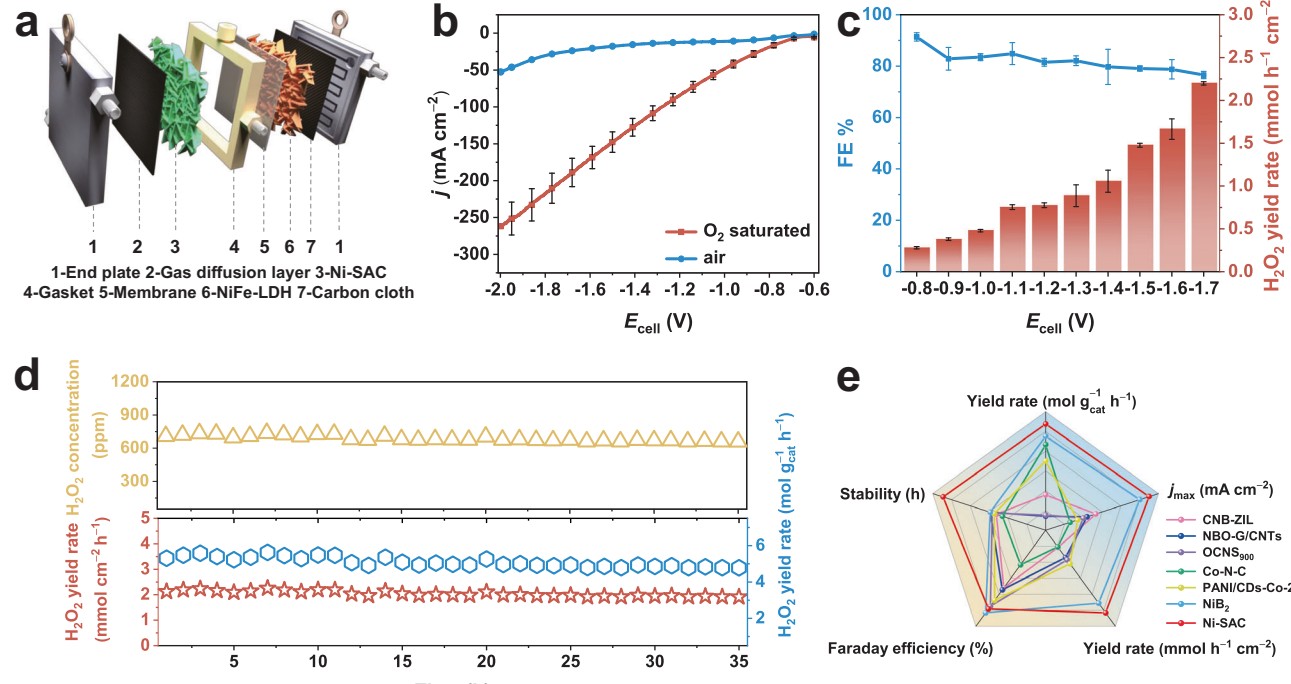

**Fig. 3 | Properties of the Ni-SAC catalyst in a two-electrode flow cell.**
**a** Schematic diagram of the flow cell. **b** ORR polarization curves in the flow cell. **c** FE and yield rate of $H_2O_2$ in the voltage range of −0.8 V to −1.7 V. **d** Stability test of Ni-SAC at −1.7 V for 35 h. **e** Comparison of 2e$^-$ ORR performances with reported electrocatalysts in the flow cell. The two-electrode system constructed in the flow cell has no *iR*-compensation. The error bars are defined as standard deviation, and the centre of each error bar represents the mean value of the corresponding three independent experiments.

in enhancing the 2e$^-$ ORR selectivity. RRDE measurement for the CNS shows an *n* value close to 4, indicating that the CNS substrate is not the reactive centre for the 2e$^-$ ORR to produce $H_2O_2$. The calculated turnover frequency (TOF) values of Ni-SAC are 0.65 s$^{-1}$, 0.72 s$^{-1}$, and 0.78 s$^{-1}$, corresponding to 0.7 V, 0.65 V, and 0.6 V, respectively. Additionally, the mass activity of Ni-SAC at 0.65 V is 49.06 A g$^{-1}$. These indices representing intrinsic activity are superior to most reported values. (Supplementary Fig. 27, Supplementary Tables 6 and 7, specific formulae are provided in Supplementary Note 1).

We also tested the reaction activity of the $H_2O_2$RR in $N_2$-saturated 0.1 M KOH with 10 mM $H_2O_2$ solution to compare the performance of the 2e$^-$ ORR over different catalysts. Figure 2g shows that the $H_2O_2$RR reactivity of Ni-SAC is significantly weaker than its 2e$^-$ ORR activity, indicating that $H_2O_2$ is relatively stable and can accumulate over Ni-SAC under reaction conditions. In contrast, the $H_2O_2$RR activity of the Ni-CNS and CNS catalysts is only slightly weaker than their 2e$^-$ ORR activity, leading to more pronounced 4e$^-$ ORR competition. To further demonstrate that the single Ni atom is the primary active site for the 2e$^-$ ORR to $H_2O_2$, we used 0.1 M KOH with the addition of 1 mM thiocyanate ion (SCN$^-$) as the toxicant to block single Ni atoms. As shown in Fig. 2h and Supplementary Fig. 28, a significant current decay (e.g., 18.03 mA cm$^{-2}$ at 0.4 V vs. RHE) is observed over Ni-SAC after adding SCN$^-$ to the electrolyte, and the current decay of the Ni-CNS decreases by only 4.51 mA cm$^{-2}$ under the same conditions. Moreover, the *n* of the poisoned Ni-SAC was determined by the RRDE to be approximately 4, echoing the *n* of the CNS (Fig. 2i and Supplementary Fig. 29). The above results strongly indicate that the single-atom Ni is the principal active site for enhancing the 2e$^-$ ORR performance.

## Electrochemical ORR to produce $H_2O_2$ in a flow cell

Motivated by the excellent 2e$^-$ ORR performance of the three-electrode system, we further evaluated the catalytic performance of the Ni-SAC electrosynthesis of $H_2O_2$ in a more practical scenario. Specifically, as displayed in Fig. 3a, we carried out two-electrode tests

in a custom two-electrode flow cell using Ni-SAC as the cathode and NiFe-LDH grown on carbon cloth as the anode. NiFe-LDH has been shown to be one of the most efficient catalysts for the oxygen evolution reaction (OER)[41]. The flow cell was equipped with a Nafion 117 membrane, and the reaction was performed in 1 M $O_2$-saturated KOH at room temperature. As shown in Fig. 3b, LSV tests were performed over a potential interval from −0.8 V to −2 V. The constructed system delivers a high current density of −261.73 ± 22.07 mA cm$^{-2}$ at a cell voltage of −2 V. We then carried out potentiostatic measurements to evaluate the FE and productivity of $H_2O_2$ in the cathodic chamber at different cell voltages. As shown in Fig. 3c, the $H_2O_2$ yield rate gradually increases as the reaction potential becomes negative, rising from 0.28 ± 0.01 mmol h$^{-1}$ cm$^{-2}$ to 2.19 ± 0.02 mmol h$^{-1}$ cm$^{-2}$ when the cell voltage increases from −0.8 V to −1.7 V. The FE of $H_2O_2$ reaches up to 91.36 ± 1.61% and maintains approximately 80% in a wide potential window. A stability test was conducted for 35 h at a cell voltage of −1.7 V, during which the concentration of generated $H_2O_2$ remained essentially stable (Fig. 3d). We then performed a comprehensive performance comparison of reported electrocatalysts for the 2e$^-$ ORR considering factors such as FE, yield rate, stability, and $j_{max}$. As shown in Fig. 3e and Supplementary Table 8, the developed Ni-SAC has superior $H_2O_2$ productivity (up to 5.48 mol g$_{cat}^{-1}$ h$^{-1}$) and can steadily operate even at high potentials. Furthermore, we calculated the electron consumption rate ($R_e$) of the constructed systems[42], and the maximum $R_e$ value achievable in the flow cell is approximately 7.56 times greater than that in the H-cell (28.52 e$^-$ s$^{-1}$ vs. 3.76 e$^-$ s$^{-1}$) (Supplementary Fig. 30).

## Mechanistic studies

The 2e$^-$ ORR consists of two proton-coupled electron transfer processes in which *OOH serves as the key reactive intermediate. In situ Fourier transform infrared spectrometry (FTIR) was carried out to monitor the adsorbed *OOH intermediate over different catalysts (including Ni-SAS, Ni-CNS, and CNS) during the ORR process to

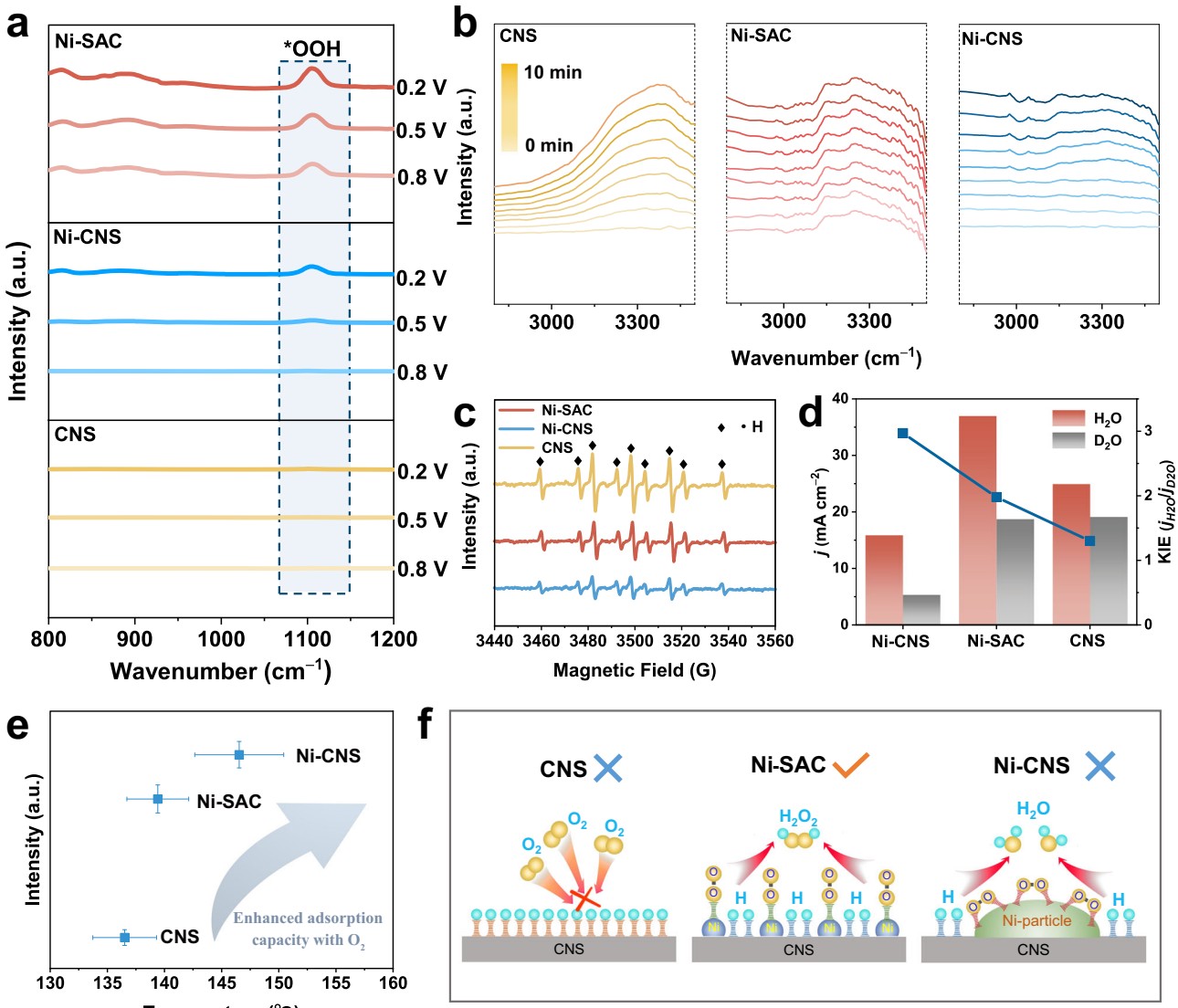

**Fig. 4 | Active hydrogen and oxygen adsorption jointly promote the 2e⁻ ORR.** **a** In situ FTIR spectra of Ni-SAC, Ni-CNS, CNS at potential of 0.2 V, 0.5 V, 0.8 V in 0.1 M O₂-saturated KOH. **b** In situ FTIR spectra of Ni-SAC, Ni-CNS, CNS at potential of 0 V in 0.1 M O₂-saturated KOH for 10 min. **c** ESR spectra of Ni-SAC, Ni-CNS, CNS. **d** KIE of H/D and current densities over Ni-SAC, Ni-CNS and CNS at 0 V. **e** O₂-TPD results for the Ni-SAC, Ni-CNS, CNS. **f** Mechanism diagram. The error bars are defined as standard deviation, and the centre of each error bar represents the mean value of the corresponding three independent experiments.

understand the reaction mechanism. As shown in Fig. 4a and Supplementary Fig. 31, the FTIR spectra display a characteristic peak at 1100 cm⁻¹, representing the stretching of O−O bonds in the adsorbed *OOH on the electrode surface[43,44]. The Ni-SAC exhibits stronger *OOH adsorption than the Ni-CNS and CNS, indicating a greater 2e⁻ ORR preference. The peak intensity of *OOH over Ni-SAC gradually increases from 0.8 to 0.2 V vs. RHE due to the accumulation of the *OOH intermediate, thus attaining higher H₂O₂ productivity (consistency with the results in Fig. 3c). The intensity of *OOH adsorption in the Ni-CNS is weak, with evident peak vibrations occurring only at higher overpotentials. The CNS shows almost no *OOH vibration at the test potentials, demonstrating poor H₂O₂ production ability. Focusing on the steps of *OOH generation (i: $O_2 + * \rightarrow *O_2$; ii: $*O_2 + H_2O + e^- \rightarrow *OOH + OH^-$), *OOH formation is related not only to suitable adsorption between the electrocatalyst and O₂ but also to the continuous and efficient supply of active hydrogen (H*) from water splitting.

We employed in situ FTIR as well as electron spin resonance (ESR) to determine the capacity for H* production, followed by the kinetic isotope effect (KIE) to assess the proton transfer capacity. According to previous studies, the vibrations appearing in the 3000−3700 cm⁻¹ band can be attributed to the stretching of O−H in H₂O[45]. As shown in Fig. 4b, the peak intensity of the O−H vibration over the CNS is stronger than that over the Ni-SAC and Ni-CNS under the same electrolytic conditions, indicating a greater water splitting ability for the CNS sample. We then conducted ESR to monitor the amount of H* during the ORR process using 5,5-dimethyl-1-pyrroline-N-oxide (DMPO) as an H* capturing agent. The classical nonuple peak intensity ratio (1:1:2:1:2:1:2:1:1) and the hyperfine coupling constants values ($A_N = 16.5$ G and $A_{H\beta} = 22.5$ G) both indicate the presence of DMPO·H, confirming H* aggregation in the three electrocatalysts(Fig. 4c and Supplementary Fig. 32; for detailed discussion, see Supplementary Note 2)[46,47]. The highest peak intensity observed in CNS can be attributed to its strong water splitting capacity, thereby maximizing the accumulation of H*. Based on this phenomenon, we determined the KIE of H/D (H₂O/D₂O) to rationalize the kinetic significance of reactive hydrogen (H*) production from the water splitting. The KIE value, representing the electron transfer rate, is determined by comparing the current densities obtained from LSV

curves scanned in 0.1 M $O_2$-saturated KOH/KOD solutions. A KIE value close to 1 indicates an acceleration of the water splitting process during hydrogenation[48,49]. Figure 4d and Supplementary Fig. 33 display the varying degrees of attenuation in catalyst reactivity following the replacement of $H_2O$ and KOH with $D_2O$ and KOD. The KIE values of Ni-SAC, Ni-CNS, and CNS are 1.98, 2.97, and 1.28, respectively. Among them, CNS has the highest proton transfer rate, followed by Ni-SAC and Ni-CNS. On the basis of the above experimental results, CNS is found to be the main site of water splitting for H* production. However, the 2e⁻ ORR performance does not positively correlate with the H* production capacity, indicating that it is not the only factor contributing to the performance improvement.

In addition to H* generation, the adsorption of $O_2$ on the catalytic surface (configuration, capacity) is another issue that needs to be considered. The geometrical configuration of the catalyst affects its adsorption configuration with respect to $O_2$. Atomically dispersed metal sites tend to absorb $O_2$ via an end-on configuration, which is not conducive to O−O breakage in *OOH; however, continuous metal nanoparticles prefer to absorb $O_2$ via a side-on configuration, thus favouring O−O breakage[23,50,51]. Therefore, ensuring the efficient conversion of *OOH through the construction of dispersed sites is crucial for enhancing the performance of the 2e⁻ ORR relative to that at continuous sites. We then conducted $O_2$ temperature-programmed desorption ($O_2$-TPD) to further investigate the difference in adsorption capacity between the three electrocatalysts and $O_2$. The peaks shown in Fig. 4e indicate the release of chemisorbed $O_2$ from the different catalysts. The desorption peaks for Ni-CNS, Ni-SAC, and CNS are located at 146.54 ± 3.89 °C, 139.41 ± 2.70 °C, and 136.52 ± 2.79 °C, respectively. A higher desorption temperature corresponds to an increase in the adsorption strength. Notably, Ni-CNS exhibits the highest adsorption strength for $O_2$, while Ni-SAC closely resembles CNS. Similar to previous studies indicating that Ni metal possesses excellent adsorption capacity for $O_2$, thereby promoting the oxygen dissociation processes (*OOH dissociation to *OH and *O)[33,52,53]. Although the desorption temperatures of Ni-SAC and CNS are similar, the desorption peak intensity of CNS is significantly lower compared to that of Ni-SAC (normalized by Brunauer-Emmett-Teller (BET) results to exclude the effect of pore size; Supplementary Fig. 34). The results reflect the variations in the quantity of $O_2$ desorbed over different electrocatalysts. Consequently, CNS is more difficult to adsorb $O_2$ compared to Ni-SAC.

In situ FTIR, ESR, and KIE experiments confirmed that the CNS is the main site at which water splitting occurs to produce H*. Moreover, previous studies have confirmed that dispersed atomic sites are able to adsorb $O_2$ via an end-on configuration, thus converging to the 2-electron path. The $O_2$-TPD results also show that Ni-SAC has a suitable adsorption capacity for $O_2$. Based on the above findings, the successful integration of the excellent H* generation ability for the CNS and the appropriate adsorption of $O_2$ on a single Ni atom can effectively enhance the activity of the 2e⁻ ORR for Ni-SAC (Fig. 4f).

### 2e⁻ ORR coupled with ethylene glycol oxidation

The cathodic ORR is typically coupled with the anodic OER, which is kinetically sluggish and generates low-market-value $O_2$[54]. To further reduce the overall energy consumption of the system, we employed a thermodynamically favourable small organic molecule oxidation reaction to replace the original OER at the anode (Fig. 5a). Note that the onset potential of ethylene glycol oxidation (EOR) is < 1 V, which is lower than that of the OER (1.23 V)[29]. Thus, coupling the ORR with the EOR (denoted as ORR||EOR) is more easily driven relative to the ORR||OER system, which contributes to lowering the cell voltage (Fig. 5b). Moreover, given the high annual consumption of poly(ethylene terephthalate) (PET) plastics and their negligible recycling, ethylene glycol (EG) derived from PET depolymerization is an excellent candidate as a key monomer for small organic molecules[55,56]. Glycolic acid (GA) prepared by the selective oxidation of ethylene glycol is also notable for its high value-added properties. We adopted Au/Ni(OH)₂, which has been reported by our group, as the electrocatalyst to evaluate the EOR performance in an electrolyte configuration of 1 M KOH + 0.3 M EG[57]. The morphology and structure of Au/Ni(OH)₂ are described in detail in Supplementary Fig. 35.

We first investigated the EOR performance of Au/Ni(OH)₂ in a three-electrode system, achieving a FE and yield rate of GA up to 92% and 1.85 mmol h⁻¹ cm⁻², respectively (Supplementary Figs. 36 and 37). We then evaluated the coupled system performance in a custom two-electrode flow cell with Ni-SAC as the cathode and Au/Ni(OH)₂ as the anode. As shown in Fig. 5c, the ORR||EOR shows the lowest cell voltage and the highest current density compared to the ORR||OER and HER||EOR systems. Quantitative analysis of the cathode and anode products over a wide potential range of −0.3 V to −1.7 V shows that the FEs of both products primarily maintain above 90%. The FE and yield rate for $H_2O_2$ reach up to 99.83% and 2.92 mmol h⁻¹ cm⁻² (7.3 mol $g_{cat}^{-1}$ h⁻¹), respectively, while the FE and yield rate for GA reach up to 97.80% and 2.09 mmol h⁻¹ cm⁻², respectively (Fig. 5d). The stability tests of the coupling system were conducted for 10 h at a cell voltage of −1.7 V (Supplementary Fig. 38). Moreover, compared with that of the ORR||OER (NiFe-LDH as the anode), the $H_2O_2$ productivity of the ORR||EOR is also improved (Supplementary Fig. 39). The ORR||EOR system can produce more $H_2O_2$ per 1 kWh of electricity at −1.7 V relative to ORR||OER systems (0.240 g vs. 0.226 g; Fig. 5e). All above results show that coupling enhance the performance of the system.

Then, we converted the alkaline electrolyte containing $H_2O_2$ directly into the downstream product, sodium perborate (SPB), to reduce the separation cost. SPB is also an oxidant that can be used for water treatment. The specific process is shown in Supplementary Fig. 40. Sodium metaborate was reacted with the electrolyte containing $H_2O_2$ in an ice-water bath for 1 h, followed by filtration and drying to obtain SPB. The XRD patterns and FTIR spectra confirm the successful preparation of the SPB products (Fig. 5f and Supplementary Fig. 41). In addition, techno-economic evaluations of the ORR||EOR and ORR||OER systems were carried out to further validate the potential of the constructed systems for industrial application as well as the superiority of the coupled systems. We evaluated the input costs in three broad categories: input chemicals, capital, and operations. The product revenues were then considered to determine the gross profits from the above two systems (the specific calculation process is detailed in Supplementary Note 3). Notably, the ORR||EOR system generates higher revenue from $H_2O_2$ production relative to the ORR||OER (7.06*10⁶ $ vs. 5.6*10⁶ $), consistent with the quantitative results. Additionally, the corresponding anode product, glycolic acid, has an extraordinarily higher value added relative to oxygen (2.83 $/kg vs. 0.09 $/kg), resulting in a substantial increase in the overall profit (15.65*10⁶ $ vs. 2.7*10⁶ $) (Fig. 5g and Supplementary Table 9).

## Discussion

In summary, we constructed a Ni-SAC hierarchical electrode with dual active sites for the efficient production of $H_2O_2$ by the 2e⁻ ORR under alkaline conditions, achieving a high yield rate of 5.48 mol $g_{cat}^{-1}$ h⁻¹ at industrial current. Based on the experimental results, Ni-SAC demonstrates the strongest *OOH adsorption since the CNS provides abundant H* combined with the suitable oxygen adsorption of single Ni atoms, synergistically enhancing the 2e⁻ ORR performance. Energy consumption is further reduced by coupling the glycol oxidation reaction, resulting in an $H_2O_2$ yield rate of up to 7.30 mol $g_{cat}^{-1}$ h⁻¹. The electrolyte containing $H_2O_2$ can be directly converted to a downstream product to reduce separation costs. This work represents a promising and energy-saving design for the alkaline electrosynthesis of $H_2O_2$ with potential applications.

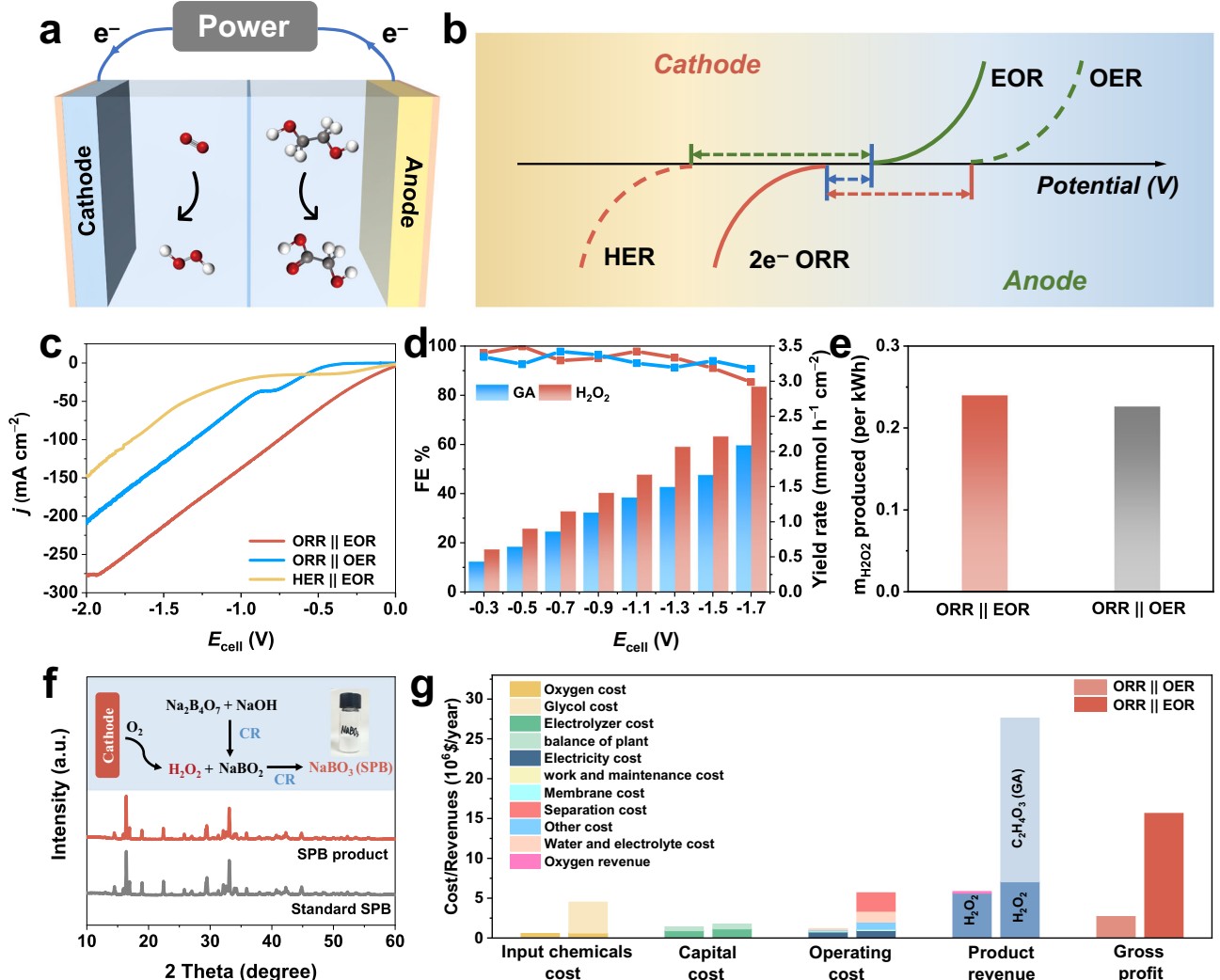

**Fig. 5 | ORR∥EOR coupling system performance and economic feasibility analysis. a** Coupling system construction diagram. **b** Comparison diagram of cell voltage for different coupling system. **c** Polarization curves in ORR∥EOR, ORR∥OER and HER∥EOR system in the flow cell. **d** FE and yield rate of $H_2O_2$ and GA in ORR∥EOR system. **e** Mass of $H_2O_2$ produced per kWh of electricity at −1.7 V in ORR∥EOR and ORR∥OER systems. **f** XRD pattern of SPB product. **g** Techno-economic evaluation of ORR∥EOR and ORR∥OER systems (left column is ORR∥OER, right column is ORR∥EOR).

## Methods

### Chemicals

Nickel nitrate hexahydrate, aluminum nitrate, iron nitrate non-ahydrate, boric acid, hexamethylenetetramine, ammonium fluoride, metanilic acid, ethylene glycol, sodium metaborate, and potassium thiocyanate, hydrogen tetrachloroaurate trihydrate were purchased from Aladdin. Cerium sulphate was purchased from Sigma-Aldrich. All chemicals are analytical grade and used without further purification. The HCP330N carbon cloth was purchased from Shanghai Hesen Electric Co., LTD.

### Material synthesis

**Preparation of NiAl-LDH(MA).** The strategy for the fabrication of integrated electrodes by the proposed confined synthesis method stems from the previous work of our group. Specifically, we used the one-step hydrothermal method to in situ grow NiAl·LDH intercalated with organic molecules (MA) on the carbon cloth. The specific synthesis steps are as follows: 3 mmol of $Ni(NO_3)_2·6H_2O$, 1 mmol of $Al(NO_3)_3·9H_2O$, 10 mmol of hexamethylenetetramine and 8 mmol of $NH_4F$ were dissolved in 25 mL of deaerated $H_2O$ to form a transparent green solution (solution A). In addition, 10 mmol MA was dissolved in

25 mL of deaerated $H_2O$ (solution B). After the two solutions were stirred evenly, solution A was added to solution B under the protection of nitrogen atmosphere and stirred until evenly dispersed. The resulting mixture was then placed in a stainless-steel high-pressure reactor lined with Teflon, and then put a piece of carbon cloth (30*50 mm²) in it. The reactor was sealed and reacted at 100 °C for 6 h. At the end of heating, cooled naturally to room temperature before removing, rinsed with deionized water and ethanol, respectively, and dried in an oven at 60 °C overnight.

**Preparation of Ni-CNS, Ni-SAC, and CNS.** The NiAl-LDH(MA) was placed on a porcelain boat and then transferred to the tube furnace at a programmable temperature. The parameters were set as follows: the temperature was kept at 800 °C for 2 h under nitrogen atmosphere, and the heating rate was 2 °C min⁻¹. After natural cooling to room temperature, the samples were removed, rinsed with deionized water and ethanol solution respectively, and then dried at 60 °C overnight. The obtained sample was denoted as Ni-CNS. Subsequently, the sample was acid etched in 1 M HCl for 12 h, then rinsed with deionized water and ethanol, respectively, and dried to obtain Ni-SAC. The CNS were obtained by heat etching the Ni-CNS in 1 M HCl for 12 h at 60 °C.

**Preparation of NiFe-LDH.** Nickel foam was pretreated with 2 M HCl, acetone, ethanol, and deionized water by ultrasonic treatment in order to remove the surface oxides and impurities. NiFe-LDH was prepared by in situ growth on pretreated nickel foam by electrodeposition at −1 V for 100 s in an electrolyte configuration of 0.15 M $Ni(NO_3)_2·6H_2O$ + 0.15 M $Fe(NO_3)_2·9H_2O$ using saturated calomel electrodes and Pt foil as reference and counter electrodes, respectively.

**Preparation of Au/Ni(OH)$_2$.** The $Ni(OH)_2$ was grown in situ on the pretreated nickel foam substrate by electrodeposition in 0.3 M $Ni(NO_3)_2·6H_2O$ solution at −1.2 V for 300 s using the saturated calomel electrode and Pt foil as the reference and counter electrode, respectively. Then, we use Ag/AgCl and the Pt foil as the reference and counter electrodes in the solution configuration of 10 mM $HAuCl_4$ + 0.5 M $H_3BO_3$ at −1 V for 600 s to finally obtain Au/Ni(OH)$_2$.

## Characterizations

Scanning electron microscopy (SEM) images were acquired from the Zeiss SUPRA 55 with an acceleration voltage set at 20 kV. High-resolution transmission electron microscopy images were performed using a JEOL-2100F coupled to an Oxford X-max EDX device with an acceleration voltage of 200 kV. The XRD patterns were obtained by the Shimadzu XRD-600 diffractometer using a Cu Kα source with a scanning step of 10 °C min$^{-1}$ and a scanning range of 3° – 70°. X-ray photoelectron spectroscopy (XPS) measurements were performed on Thermo VG ESCALAB 250 using the Al Kα source at a pressure of approximately 2*10$^{-9}$ Pa. All the binding energies were calibrated with the position of 284.6 eV corresponding to the adventitious carbon. The fitting of spectral peaks in the XPS data was performed using the XPSPEAK41 software, which applied a mixed Gaussian-Lorentzian function. The atomic-resolution high angle annular dark-field scanning transmission electron microscopy images (HAADF-STEM) were recorded on the JEOL JEM-ARM200F TEM/STEM operating at 300 kV with a spherical aberration correction. The UV-Vis spectra were acquired from Shimadzu UV-2600. The XAS spectra were measured on a hard X-ray spectrometer at the beamlines of the Shanghai Synchrotron Radiation Facility (SSRF) and Beijing Synchrotron Radiation Facility (BSRF). Extended X-ray absorption fine structure spectra (EXAFS) were recorded at ambient temperature in fluorescence mode and transformed without phase correction. The O$_2$-TPD spectra were collected on the chemisorption analytical instrument of Micromeritics Auto-Chem II 2920. In situ FTIR measurements were realizing by combining the TENSOR∥FTIR with CHI760E electrochemical workstation. Electron spin resonance (ESR) measurements were implemented using the Bruker ECSEMX X-band ESR spectrometer at room temperature.

## Electrochemical measurement

All electrochemical measurements were performed on the CHI760E workstation without any *iR*-compensation at room temperature. A graphite rod and a Ag/AgCl (saturated KCl) were used as the counter electrode and the reference electrode. All the potentials were converted to reference the reversible hydrogen electrode (RHE) by $E_{RHE} = E_{Ag/AgCl} + 0.197 + 0.059 \times pH$. The calibration details of the Ag/AgCl vs. RHE are constructed as a three-electrode system under H$_2$-saturated conditions with a Pt wire and a graphite rod as the working electrode and counter electrode, respectively. Cyclic voltammetry (CV) tests were carried out in this system at a scan rate of 1 mV s$^{-1}$, and the average of the two potentials at which the curve crosses the current to zero was taken as the calibration value.

The freshly prepared 0.1 M KOH electrolyte (pH = 13) should be introduced into O$_2$/N$_2$ for 30 min in advance to achieve the saturation of the atmosphere according to the experimental requirements. Electron transfer number (*n*) and HO$_2^-$ % were determined using the rotating ring disk electrode (RRDE) as the working electrode (work area: 0.1256 cm$^2$). 4 mg of the catalyst powder was dispersed into

770 μL of water, 200 μL of ethanol, and 30 μL of 5 wt% Nafion. Ultrasonication was performed to obtain a uniformly dispersed black ink. 4.4 μL of the suspension was dropped onto the polished electrode surface and dried at room temperature. The catalyst was scanned by CV over the potential range of 1.2 V to 0 V vs. RHE at a scan rate of 50 mV s$^{-1}$ up to steady state. LSV tests in the RRDE set-up were obtained at a potential range of 1 V to 0 V vs. RHE with a scan rate of 5 mV s$^{-1}$ and a rotating speed of 1600 r.p.m., while its ring voltage was set to 1.2 V vs. RHE. Correcting ORR polarization curves by deducting the current response measured at N$_2$-saturated condition. The *n* and HO$_2^-$ % are calculated by the following equation (Eqs. 1 and 2):

$$n = 4 \times \frac{I_d}{I_d + I_r/N} \tag{1}$$

$$HO_2^- \% = 200 \times \frac{I_r/N}{I_d + I_r/N} \tag{2}$$

where $I_r$ is the ring current, $I_d$ is the disk current, $N$ is the collection efficiency (0.37 after calibration).

For the RDE measurements, the polarization curves were carried out at 400 rpm, 625 rpm, 900 rpm, 1225 rpm, 1600 rpm and 2025 rpm with the speed of 5 mV s$^{-1}$, respectively, and the number of electron transfer can be calculated by the Koutecky–Levich (K – L) equation (Eqs. 3–5):

$$J^{-1} = J_L^{-1} + J_K^{-1} = B^{-1}w^{-1/2} + J_K^{-1} \tag{3}$$

$$B = 0.62nFC_0D_0^{2/3}v^{-1/6} \tag{4}$$

$$J_K = nFkC_0 \tag{5}$$

where $J$ is the measured current density; $J_K$ and $J_L$ are the kinetic- and diffusion-limiting current density; ω is the electrode rotating rate; $F$ is the Faraday constant ($F = 96485$ C mol$^{-1}$); $C_O$ is the bulk concentration of O$_2$ (1.2*10$^{-6}$ mol cm$^{-3}$); $D_O$ is the diffusion coefficient of O$_2$ in 0.1 M KOH solution (1.9*10$^{-5}$ cm$^2$ s$^{-1}$); ν is the kinematic viscosity of the electrolyte (0.01 cm$^2$ s$^{-1}$), and $k$ is the electron transfer rate constant.

Faraday efficiency (FE) and yield rate were measured using the integrated electrode as the working electrode (loading: 0.4 mg cm$^2$) in the H-cell with Nafion 117 as the separator (thicknesses: 183 μm, size: 4 × 4 cm$^2$, Dupont). Pretreatment of Nafion 117 membrane was carried out by treating with 5% mass fraction of hydrogen peroxide at 80 °C for 1 h, followed by soaking in deionized water for 30 min, and then treating with 5% mass fraction of H$_2$SO$_4$ at 80 °C for 1 h, and finally soaking in deionized water for 30 min. The area of the working electrode was regulated at 0.5 cm$^2$. Anode and cathode compartments were all filled with 15 mL of 0.1 M KOH (the difference is that the cathode is O$_2$-saturated), respectively. The potentiostatic method was employed to evaluate the corresponding FE and H$_2$O$_2$ yield rate by testing at different voltages (from 0.7 V to 0.2 V vs. RHE with 0.1 V per potential interval) for 1 h. The FE calculation formula is shown below (Eq. 6):

$$FE = \frac{2cVF}{Q} \tag{6}$$

where $F$ is the Faraday constant ($F = 96485$ C mol$^{-1}$), $c$ is the H$_2$O$_2$ concentration, $V$ is the electrolyte volume, $Q$ is the total charge.

According to the previously reported work, we employed the cerium sulfate (Ce(SO$_4$)$_2$) method to quantify the production capacity of H$_2$O$_2$. Ce(SO$_4$)$_2$ was configured as 0.1-0.5 mM Ce(SO$_4$)$_2$ using 0.5 M H$_2$SO$_4$ as the solvent, and the standard curve between Ce$^{4+}$ concentration and absorbance was subsequently established by

measuring the absorbance at 319 nm. The cathode electrolyte obtained after potentiostatic test was added to 0.5 mM $Ce(SO_4)_2$, and the change in $Ce^{4+}$ concentration before and after the reaction was monitored using the UV-Vis spectrophotometer (Since yellow $Ce^{4+}$ reacts with $H_2O_2$ to form colorless $Ce^{3+}$, Eq. 7), after which the concentration of $H_2O_2$ could be calculated by stoichiometric relationship ($H_2O_2$ concentration was half of the consumed $Ce^{4+}$ concentration).

$$2Ce^{4+} + H_2O_2 \rightarrow 2Ce^{3+} + 2H^+ + O_2 \qquad (7)$$

The quantitative results were averaged after three parallel experiments and the standard deviation was calculated as follows (Eq. 8):

$$\sigma = \sqrt{\frac{\sum_{i=1}^{n}(x_i - \bar{x})^2}{n-1}} \qquad (8)$$

where $\sigma$ is standard deviation, $x_i$ is the measured data of products, $\bar{x}$ is the average value of measurements data, and $n$ is experimental repetition times.

Stability tests were operated under $O_2$-saturated 0.1 M KOH at 0.2 V vs. RHE. The $H_2O_2RR$ test was performed by LSV scanning under the $N_2$-saturated atmosphere with an electrolyte of 0.1 M KOH + 10 mM $H_2O_2$. Poisoning experiments were carried out using 0.1 M KOH + 1 mM KSCN electrolyte configuration under the $O_2$-saturated atmosphere for LSV and RRDE tests. KIE was tested for LSV measurements at 0.1 M $O_2$-saturated KOD (KOD and $D_2O$ instead of KOH and $H_2O$). The resistance measurement was performed in the frequency from range from 0.1 Hz to 100000 Hz.

Electrochemically active surface area (ECSA) measured by double layer capacitance method. In the non-Faradaic interval, the cyclic voltammetry (CV) tests were implemented at the scan rates of 20, 40, 60, 80, and 100 mV s$^{-1}$ in 0.1 M $N_2$-saturated KOH to exclude the effect of trace oxygen in the electrolyte. Plotting $\Delta j$ as a function of scan rate, the slope of which is the double layer capacitance ($C_{dl}$).

The membranes assembled in the two two-electrode flow cell devices mentioned below are consistent with that employed in the H-cell, with an electrolyte volume of 50 mL in each compartment. The working electrode area of the catalyst is 1 cm$^2$. The two-electrode uncoupled system flow cell was assembled with the developed Ni-SAC as the cathode and NiFe-LDH as the anode for the test in 1 M KOH. The potentiostatic test is performed in the potential range from −0.8 V to −1.7 V, with a 0.1 V interval between each potential. For the coupled system, Au/Ni(OH)$_2$ was used as the anode and the solution configuration was replaced by 1 M KOH + 0.3 EG, while other conditions remained unchanged. Potentiostatic tests are implemented from −0.3 V to −1.7 V, with an interval of 0.2 V.

### In situ FTIR experiments
The TENSOR II FTIR was coupled to a CHI 760E electrochemical workstation in a custom-made single cell for testing in the three-electrode system using potentiostatic mode. The infrared spectrum collected at the open circuit voltage was used as the background spectrum, and the actual result obtained was the spectrum after subtracting the background spectrum.

### TPD measurements
Weigh 50–100 mg of ground sample in the reaction tube, warm up from room temperature to 300 °C at 10 °C min$^{-1}$ for drying pretreatment, purge with He air flow (30–50 mL min$^{-1}$) for 1 h, cool down to 50 °C, add 10% $O_2$/He mixture (30–50 mL min$^{-1}$) for 1 h to saturation, switch to He air flow (30–50 mL min$^{-1}$) and purge for 1 h to remove the weakly adsorbed $O_2$ on the surface, and finally desorb under He atmosphere at a warming rate of 10 °C min$^{-1}$ to 300 °C. The surface was purged with He gas (30–50 mL min$^{-1}$) for 1 h to remove the weakly adsorbed $O_2$, and then finally desorbed at 300 °C under He atmosphere with a heating rate of 10 °C min$^{-1}$, and the desorbed gases were detected by TCD. The intensity is normalized by BET results.

### ESR measurements
The electron spin resonance (ESR) experiment was implemented as follows: we performed the spectra collection in potentiostatic mode at 0.3 V vs. RHE reaction for 600 s followed by the addition of 200 μL of DMPO.

## Data availability
The data supporting the findings of this study are available within the article and its Supplementary Information. The source data generated in this study are available in the figshare repository (https://doi.org/10.6084/m9.figshare.26058649). Additional data are available from the corresponding authors upon reasonable request. Source data are provided with this paper.

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

## Acknowledgements

This work was financially supported by National Natural Science Foundation of China (Grant No. 22090031, 22090030, 22288102 to M.S. and Z.L.), Young Elite Scientist Sponsorship Program by CAST (Grant No. 2021QNRC001 to Z.L.) and the Fundamental Research Funds for the Central Universities (Grant No. buctrc202011 to Z.L.). The authors were

thankful for the support of the BSRF (Beijing Synchrotron Radiation Facility) and SSRF (Shanghai Synchrotron Radiation Facility) during the XAFS measurements at the beamline of 1W1B and BL11B.

## Author contributions

Y.S. designed and carried out the synthesis, characterizations and catalytic reactions, analyzed the data, and wrote the manuscript. K.F. and J.L carried out catalytic reactions and analyzed the data. Y.Y. and L.W. analyzed and fitted the XAFS data. Z.L., M.S. and X.D. supervised the project, conceived the idea, helped design the experiments, analyzed the data, and wrote the manuscript. All the authors commented on the manuscript and have given approval for the final version of the manuscript.

## Competing interests

The authors declare no competing interests.
