## [Peer Review File · Nature Communications]

REVIEWER COMMENTS

Reviewer #1 (Remarks to the Author):

In this work, the authors reported a highly efficient 2e⁻ ORR electrocatalyst of hierarchical carbon nanosheet array electrode decorated with Ni single atoms (Ni-SAC) via the thermal pyrolysis of organic molecule-intercalated layered double hydroxides. The catalyst was fully characterized and the reaction mechanism for the ORR was studied in detail with various characterization tools such as electron spin resonance, kinetic isotope effect and in situ Fourier transform infrared spectroscopy. In addition, coupled system was constructed by further electrochemical pairing of ethylene glycol reaction to achieve performance enhancement and then complete the direct conversion of hydrogen peroxide to downstream products. This work holds significant implications for the efficient alkaline electrosynthesis of H₂O₂. Consequently, I recommend its publication in Nature Communications after the authors properly address the following issues.

1. Poisoning study of SCN⁻ should be performed on control samples to verify the unique role of nickel single atom in enhancing the catalytic performance.
2. For the two-electrode flow cell, the authors should provide longer-term stability test (Figure 3d) to demonstrate its potential viability for industrial applications.
3. The techno-economic analysis should also take into account the fluctuation of input chemicals and product prices. Accordingly, please refer to sufficient data and information to give a more objective techno-economic analysis, or at least give an error margin due to price fluctuations.
4. The intrinsic activity such as the turnover frequency and mass activity should be provided and compared with literature values.
5. The discussion regarding the structure-property correlation should be enhanced throughout the manuscript. The following references would be helpful: Nat. Commun. 2023, 14, 368; Small 2022, 18, 2103824.
6. For anodic ethylene glycol electrooxidation reaction, please give the structural and morphological characterization of the anodic catalyst and provide liquid phase data on the distribution of products of the ethylene glycol oxidation reaction.
7. The utilization of electron spin resonance (ESR) to monitor the amount of H^{*} during ORR process is an intriguing method. However, the article lacks explicit details on the specific experimental procedures. Please give the specific procedure for ESR. Moreover, the reason for designating the captured active species as H^{*} should be explained in depth.
8. For the quantitative results of Figures 2b and 2c, please give the corresponding error intervals.
9. The authors should carefully check the manuscript to avoid any editing errors. For example, CNS, which appears for the first time in the abstract, does not give its full name; Figure 4b and Figure 4c have the wrong order of figure annotations; journal name not was italicized in reference 4.

Reviewer #2 (Remarks to the Author):

In this study, Sun et al. reported on the electrochemical ORR for H₂O₂ production using a hierarchical carbon nanosheet array electrode containing Ni-SAC. The yield was measured at 0.685 mol g⁻¹ h⁻¹ and 5.475 mol g⁻¹ h⁻¹, with FE of 89% and 91% in the H-cell and the flow cell, respectively. While the presented H₂O₂ yield rate is noteworthy, the concept of employing Ni single atoms is not novel, and further material characterizations require more elaboration. Additionally, I do not see the presence of pure Ni-SAC, and the presented data actually suggests the existence of Ni nanoclusters in the matrix. In summary, considering the impressive electrochemical results, I recommend its publication in Nature Communications after a major revision. My detailed comments are as follows:

–Major comments

- (1) The enhancement in the intensity of the (002) peak after the acid leaching process (Fig. 1b) raises questions. It is not clear why the acid-leaching process will improve the crystalline order of CNS. Additionally, could the authors explain the observed difference in the crystalline order of the carbon matrix in Ni-SAC and Ni-CNS (red and blue curves in Fig. 1b)?
- (2) The HAADF-STEM image (Fig. 1e) reveals the presence of agglomerated atoms or the formation of nanoclusters. It is recommended that the authors replace this image with a higher-quality version for clearer observation.
- (3) The elemental mapping EDX image (Supplementary Fig. 8b) depicting the distribution of Ni is not convincing and may require further clarification or improvement.
- (4) The XPS data analysis requires significant revision and further clarifications. The detailed comments on the XPS are as follows:
 - (4-1) The authors should provide details on how they calibrated the binding energies of the XPS spectra.
 - (4-2) The line shapes of the peaks used to fit the XPS spectra, specifically for N 1s, S 2p, and Ni 2p, should be included in the manuscript, preferably in the Method section.
 - (4-3) Assigning the peak at 853.0 eV to Ni²⁺ rather than Ni⁰ is subjected to discuss. The authors are advised to explain this assignment.
 - (4-4) The inconsistency in the peak ratio under Ni 2p_{3/2} relative to what is fitted under Ni 2p_{1/2} needs clarification. Concerning the quantum states, i.e., 2j+1 for j=1/2 and 3/2, the relative intensity ratio of Ni 2p_{3/2}–to–Ni 2p_{1/2} should be 2 for each oxidation state.

(4-5) I suggest that the authors re-conduct the XPS analysis for both Ni-SAC and Ni-CNS samples with a wider range to obtain a more accurate baseline. Additionally, they should re-fit the data and provide information about the line shapes in the Supplementary Information.

(5) The presence of a shoulder at 2.2 Å (Ni-Ni) in the Ni-SAC sample suggests the existence of Ni nanoclusters (Fig. 1h). XAS data does not indicate the presence of pure Ni single atoms in the matrix. I additionally recommend that the authors provide a model to fit the Fourier-transformed EXAFS spectra.

(6) Since XRD cannot show the presence of nanoclusters, I recommend presenting STEM images after the stability test to examine the presence of any agglomerated Ni clusters.

(7) The observed improvement in the electrochemical activity of Ni-SAC compared to Ni-CNS could be attributed to the higher surface area of Ni-SAC, as indicated by SEM images and BET data. Therefore, I suggest that the authors measure the electrochemical surface area of these samples and include a normalizing of the current based on this factor in the Supplementary Information.

(8) I recommend that the authors calculate and include the number of consumed electrons per active site per second versus E, based on the method proposed in Nat Commun 13, 1256 (2022), for both H-cell and the flow cell in the Supplementary Information, with a corresponding note in the main manuscript.

–Minor comments

(1) CNS should be defined in both the Abstract (Line 18) and Introduction (Line 72) sections.

(2) The sentence 'Pure carbon nanosheet ... condition for 12 h (Supplementary Fig. 3)' lacks clarity and needs to be rewritten for better understanding.

(3) Orbital symbols should be written in italic font.

(4) The authors are advised to plot the raw data of the XPS spectra (Fig. 1f, Supplementary Fig. 11, Supplementary Fig. 12, and Supplementary Fig. 13) with a thicker line to facilitate the examination of noise and fluctuations.

(5) The N 2s spectrum of Ni-CNS is missing. The authors should include the N 2s spectrum of Ni-CNS in the Supplementary Information.

(6) Considering the variation in loaded mass across different studies, I recommend the authors plot H₂O₂ yield (mmol h⁻¹ cm⁻²) versus E, incorporating data from other studies in the Supplementary Information, similar to Fig. 2d.

(7) Could the authors provide the error associated with the desorption peak in the O₂-TPD results (Fig. 4e)?

We appreciate the reviewers for their valuable comments; a series of new experiments have been performed to enable us to fully address the comments raised by the reviewers. Details of the corresponding revisions are described below point by point.

Response to Reviewer 1#

In this work, the authors reported a highly efficient $2e^-$ ORR electrocatalyst of hierarchical carbon nanosheet array electrode decorated with Ni single atoms (Ni-SAC) via the thermal pyrolysis of organic molecule-intercalated layered double hydroxides. The catalyst was fully characterized and the reaction mechanism for the ORR was studied in detail with various characterization tools such as electron spin resonance, kinetic isotope effect and in situ Fourier transform infrared spectroscopy. In addition, coupled system was constructed by further electrochemical pairing of ethylene glycol reaction to achieve performance enhancement and then complete the direct conversion of hydrogen peroxide to downstream products. This work holds significant implications for the efficient alkaline electrosynthesis of H_2O_2 . Consequently, I recommend its publication in Nature Communications after the authors properly address the following issues.

Comment 1. Poisoning study of SCN^- should be performed on control samples to verify the unique role of nickel single atom in enhancing the catalytic performance.

Response: Thank you for this valuable suggestion. In accordance with the reviewer's comments, we have supplemented the poisoning study of the Ni-CNS to further demonstrate the important role of nickel single atoms in the enhancement of reactivity. The experiments were implemented by using the same electrolyte configuration (0.1 M KOH + 0.001 M KSN) as in the poisoning Ni-SAC experiments (original Fig. 2h). The current density of Ni-SAC decayed by 18.03 mA cm^{-2} at 0.4 V vs. RHE after the poisoning experiment, whereas Ni-CNS only decayed by 4.51 mA cm^{-2} under the same

conditions, reflecting the important role of Ni single atoms in enhancing the reaction activity (**Revised Supplementary Fig. 28**). The corresponding discussion has been presented in the revised manuscript, as the following: “As shown in Fig. 2h, a significant current decay (e.g., 18.03 mA cm⁻² at 0.4 V vs. RHE) is observed over Ni-SAC after adding SCN⁻ in the electrolyte, whereas Ni-CNS only decayed by 4.51 mA cm⁻² under the same conditions (Supplementary Fig. 28). Moreover, the *n* of the poisoned Ni-SAC was determined by RRDE to be about 4, echoing with the *n* of CNS (Fig. 2i and Supplementary Fig. 29). The above results strongly suggest that the Ni single atom is the principal active site to enhance the 2e⁻ORR performance.” (please see line 218, Page 11 in the revised manuscript)

Revised Supplementary Fig. 28 ORR polarization curves before and after 1 mM SCN⁻ poisoning in 0.1 M O₂-saturated KOH for **a**, Ni-SAC and **b**, Ni-CNS.

Comment 2. For the two-electrode flow cell, the authors should provide longer-term stability test (Figure 3d) to demonstrate its potential viability for industrial applications.

Response: We thank the reviewer for this insightful comment. According to the comment, we have provided a longer stability test to demonstrate the potential of Ni-SAC in the future applications, which has been updated in the revised manuscript (**Revised Fig. 3d**).

Revised Fig. 3d Stability test of Ni-SAC in flow cell at -1.7 V.

Comment 3. The techno-economic analysis should also take into account the fluctuation of input chemicals and product prices. Accordingly, please refer to sufficient data and information to give a more objective techno-economic analysis, or at least give an error margin due to price fluctuations.

Response: We thank the reviewer for this insightful comment. Based on the reviewers' suggestions, we have re-evaluated the techno-economic analysis and updated the detailed calculations process as well as Fig. 5g after reviewing relevant websites and literature in view of fluctuations in raw material and product price. The total benefits of ORR || OER and ORR || EOR change from the original 3.7×10^6 \$ and 12.7×10^6 \$ to 2.7×10^6 \$ and 15.65×10^6 \$, respectively. The corresponding discussion has been presented in the revised manuscript, as the following: “Notice that ORR || EOR system has a higher revenue on the H_2O_2 product relative to ORR || OER (5.6×10^6 \$ vs. 7.06×10^6 \$), which is consistent with the quantitative results, and the corresponding anode product, ethanoic acid, has an extraordinarily high-value-added relative to oxygen (2.83 \$/kg vs. 0.092 \$/kg), resulting in a significant increase in the overall profit (2.7×10^6 \$ vs. 15.65×10^6 \$).” (please see line 376, Page 20 in the revised manuscript). Updated data and specific references are presented in the **Revised Fig. 5g and Revised Supplementary Table 9**.

Revised Fig. 5g Techno-economic evaluation of ORR || EOR and ORR || OER systems.

Revised Supplementary Table 9. Average price references and corresponding sources for raw materials and products.

Chemicals	Price (\$/kg)		Average		Source
O ₂	0.035	0.14	0.1	0.092	https://app.indexbox.io/table/280440/0/ , Nat Commun. , 2023 , 14, 6263
H ₂ O ₂	1.2	1.3	0.76	1.087	https://www.chemanalyst.com/Pricing-data/hydrogen-peroxide-1169
EG	0.56	0.73	0.48	0.59	http://www.100ppi.com
GA	2.2	2.4	3.9	2.83	https://www.pharmacompass.com/price/glycolic-acid

Comment 4. The intrinsic activity such as the turnover frequency and mass activity should be provided and compared with literature values.

Response: We sincerely thank the reviewer for the professional comment. As suggested, we further illustrate the advantages of Ni-SAC by adding two indicators that are representative of the intrinsic activity, including turnover frequency (TOF) and mass activity (MA). The formulae related to the TOF values has been illustrated in **Supplementary Note 1**. (please see line 166, Page 18 in the revised Supplementary Information)

(1) The calculated TOF values of Ni-SAC are 0.65 s⁻¹, 0.72 s⁻¹, and 0.78 s⁻¹ corresponding to 0.7 V, 0.65 V, and 0.6 V, which gives an advantageous position

over some noble metal-based, transition metal-based, and carbon-based materials (please see the Revised Supplementary Fig. 27 and Revised Supplementary Table 6 shown below). The corresponding discussion of TOF has been presented in the revised manuscript. (please see line 207, Page 11 in the revised manuscript).

Revised Supplementary Fig. 27 Comparison of TOF values of Ni-SAC with previous reported works.

Revised Supplementary Table 6. Comparison of TOF values of Ni-SAC with previous reported works.

No.	Catalyst	Electrolyte	Potential (V vs. RHE)	TOF (s ⁻¹)	Ref.
			0.7	0.0839	
1	Pt-Hg	0.1 M HClO ₄	0.65	0.19	18
			0.6	0.53	
2	Pd-Hg	0.1 M HClO ₄	0.65	0.22561	19
			0.6	0.56098	
3	Ni-N ₂ O ₂ /C	0.1 M KOH	0.65	0.02	14
4	Mo ₁ /OSG-H	0.1 M KOH	0.65	0.076	20
5	O-CNT	0.1 M KOH	0.6	0.0178	21

6	F-mrGO	0.1 M KOH	0.65	0.088	22
7	COF-366-Ni	0.1 M KOH	0.65 0.6	0.33 0.67	23
8	ZnO ₃ C	0.1 M KOH	0.65	0.197	24
9	O-C(Al)	0.1 M NaOH	0.65	0.196	25
10	200-Pt-N- CNT	0.1 M HClO ₄	0.6	0.5	26
11	Co-SCD-2	0.1 M KOH	0.6	0.044	27
12	BUCT-COF- 7/CNT	0.1 M KOH	0.7	0.053	28
			0.7	0.65	
13	Ni-SAC	0.1 M KOH	0.65	0.72	This work
			0.6	0.78	

(2) We further give the comparison of mass activity with the currently published literatures. The mass activity of Ni-SAC at 0.65 V is 49.06 A g⁻¹, which displays the impressive performance over the most reported works (**Revised Supplementary Table 7**). The corresponding discussion of mass activity has also been presented in the revised manuscript. (*please see line 208, Page 11 in the revised manuscript*).

Revised Supplementary Table 7. Comparison of mass activity of Ni-SAC with reported works.

No.	Catalyst	Electrolyte	Mass Activity (A g ⁻¹) at 0.65 V	Ref.
1	O-CNTs	0.1 M KOH	20.6	Nat. Catal. , 2018 , 1, 156

2	Co-N-C	0.1 M KOH	12.1	J. Am. Chem. Soc. , 2019 , 141, 12372
3	Ni ₂ Mo ₆ S ₈	0.1 M KOH	~ 0.93	Adv. Funct. Mater. , 2021 , 31, 2104716
4	In Sas/NSBC	0.1 M KOH	2.5	Angew. Chem. Int. Ed. , 2022 , 61, e202117347
5	OCNS	0.1 M KOH	14.5	Angew. Chem. Int. Ed. , 2021 , 60, 16607
6	Co-POC-O	0.1 M KOH	16.5	Adv. Mater. , 2019 , 31, 1808173
7	N-FLG-8	0.1 M KOH	14.65	Adv. Energy Mater. , 2020 , 10, 2000789
8	GNP _{C=0,1}	0.1 M KOH	8.87	Nat. Commun. , 2020 , 11, 2209
9	O-C(Al)	0.1 M NaOH	28.5	Nat. Commun. , 2020 , 11, 5478
10	Mo ₁ /-OSGH	0.1 M KOH	20.86	Angew. Chem. Int. Ed. , 2020 , 59, 9171
11	Ni-N ₂ O ₂ /C	0.1 M KOH	1.10	Angew. Chem. Int. Ed. , 2020 , 59, 13057
12	BUCT-COF-7/ CNT	0.1 M KOH	~11	Angew. Chem. Int. Ed. , 2023 , 62, e202314539
13	Co-SCD-2	0.1 M KOH	~26.85	Angew. Chem. Int. Ed. , 2023 , 62, e202307355
14	NBO-G/CNTs	0.1 M KOH	19.3	Adv. Mater. , 2023 , 35, 2209086
15	Ni-SAC	0.1 M KOH	49.06	This work

Comment 5. The discussion regarding the structure-property correlation should be enhanced throughout the manuscript. The following references would be helpful: *Nat. Commun.* **2023**, 14, 368; *Small* **2022**, 18, 2103824.

Response: We thank the reviewer for the valuable suggestion and providing great references. We have combined characterization and experiments to provide a more

specific explanation of the structure-property relationship, as the following: “*In situ* FTIR, ESR, and KIE experiments confirmed that CNS is the main site of water splitting to produce H*. Moreover, previous works have proven that the dispersed atomic site is able to adsorb O₂ via end-on configuration thus converging to the 2-electron path. O₂-TPD results also show that Ni-SAC has a suitable adsorption capacity with O₂. Therefore, the successful integration of the excellent H* generation ability for CNS and the appropriate adsorption of O₂ on Ni single atom for Ni-SAC can effectively enhance the activity of 2e⁻ORR.” (please see line 322, Page 17 in the revised manuscript). We have cited the relevant literature in the revised manuscript (please see the revised Ref. 23, 24, 50, 51).

Comment 6. For anodic ethylene glycol electrooxidation reaction, please give the structural and morphological characterization of the anodic catalyst and provide liquid phase data on the distribution of products of the ethylene glycol oxidation reaction.

Response: We thank the reviewer for this insightful comment. We have provided the SEM and HRTEM images of Au/Ni(OH)₂ in the **Revised Supplementary Fig. 34**. The SEM images show the Au nanoparticles were uniformly loaded on the Ni(OH)₂ nanosheets. The HRTEM results displays that the lattice fringes that can be assigned to the (101) plane of β -Ni(OH)₂ and the (111) plane of Au, respectively, affirming the successful synthesis of Au/Ni(OH)₂. The corresponding discussion has been presented in the revised supplementary information. (please see Line 127, Page 16 in the Supplementary Information)

Revised Supplementary Fig. 34 a-b SEM images of Au/Ni(OH)₂ at different magnification. c-d HRTEM images of Au/Ni(OH)₂ at different magnification.

In addition, we present the liquid phase data related to the EG electrooxidation reaction according to the reviewer's comments, thus further illustrating the production capacity of Au/Ni(OH)₂ for GA. Specifically, we provide the standard curve used to calculate the concentration of GA and identify the peaks attributed to the EG electrooxidation products in HPLC chromatogram (**Revised Supplementary Fig. 36a-b**). Subsequently, we further give the product distribution of EG electrooxidation in the potential range of 1.0-1.4 V, where GA is the main product while OA and FA are the by-products (**Revised Supplementary Fig. 36c**). The corresponding discussion has been presented in the revised Supplementary Information. (*please see Line 142, Page 17 in the Supplementary Information*)

Revised Supplementary Fig. 36 a The standard curve for glycolic acid. b HPLC

chromatogram of EG electrooxidation products over Au/Ni(OH)₂ in 1 M KOH +0.3 M EG at 1.0 V for 1 h (OA and FA represent for oxalic acid and formic acid, respectively).
c Product distribution of EG electrooxidation in the voltage range 1.0-1.4 V.

Comment 7. The utilization of electron spin resonance (ESR) to monitor the amount of H* during ORR process is an intriguing method. However, the article lacks explicit details on the specific experimental procedures. Please give the specific procedure for ESR. Moreover, the reason for designating the captured active species as H* should be explained in depth.

Response: We sincerely thank the reviewer for the professional comment.

- (1) Discussions related to electron spin resonance (ESR) operation have been presented in the revised manuscript, as the following: “we performed the spectra collection in potentiostatic mode at 0.3 V vs. RHE reaction for 600 s followed by the addition of 200 μ L of DMPO.” The corresponding discussion has been shown in the Method section of the revised manuscripts (*please see line 538, Page 28 in the Method section of revised manuscript*). Moreover, ESR test parameters were added to the characterization section of the revised manuscript, as the following: “Electron spin resonance (ESR) measurements were implemented using the Bruker ECSEM X-band ESR spectrometer at room temperature.” (*please see line 454, Page 24 in the revised manuscript*)
- (2) ESR serves as a technique for the identification of radical species by detecting relatively long-lived spin adducts (Typically nitrogen oxides, characterized by resonance stabilization attributable to their unpaired electrons) obtained by reacting radicals with diamagnetic compounds (spin traps, in this case, DMPO) (*Free Radic. Biol. Med.*, **1987**, 3, 259). The collected spectra were subsequently analyzed for hyperfine splitting (HFS) parameters as well as the magnitude of the splitting to identify the species of the captured radicals. We can identify the nonuple peaks with an intensity ratio of 1:1:2:1:2:1:2:1:1, which hyperfine coupling constants are $A_N =$

16.5 G and $A_{H\beta} = 22.5$ G (*Nat. Commun.*, **2022**, 13, 7958, *Angew. Chem. Int. Ed.*, **2022**, 61, e202209849) ((**Revised Supplementary Fig. 31**). The above results all correspond to DMPO-H adducts. The reason for designating species captured via DMPO as active hydrogen (H^*) are shown in the **Supplemental Note 2** of the revised Supplementary Information. (*please see line 180, Page 19 in the Supplementary Information*)

Revised Supplementary Fig. 31 HFS analysis of hydrogen radicals and corresponding nonuple peaks.

Comment 8. For the quantitative results of Figures 2b and 2c, please give the corresponding error intervals.

Response: We thank the reviewer for this insightful comment. We have added the error bar information in Figure 2b, 2c of the revised manuscript and updated the images and tables related to yield comparisons accordingly (*please see the Revised Fig. 2b-2c in the revised manuscript*).

Revised Fig. 2b Faradaic efficiency (FE) and **2c** H₂O₂ yield rate of each sample in the voltage range of 0.2 V-0.7 V.

The formula for calculating the standard deviation is shown below:

$$\sigma = \sqrt{\frac{\sum_{i=1}^n (x_i - \bar{x})^2}{n - 1}}$$

where σ is standard deviation, x_i is the measured data of products, \bar{x} is the average value of measurements data, and n is experimental repetition times. (please see Line 499, Page 26 in the Method section of revised manuscript)

Comment 9. The authors should carefully check the manuscript to avoid any editing errors. For example, CNS, which appears for the first time in the abstract, does not give its full name; Figure 4b and Figure 4c have the wrong order of figure annotations; journal name not was italicized in reference 4.

Response: Thank you for this valuable suggestion. We have corrected the article again and a series of editing errors have been corrected.

Response to Reviewer 2#

In this study, Sun et al. reported on the electrochemical ORR for H₂O₂ production using a hierarchical carbon nanosheet array electrode containing Ni-SAC. The yield was measured at 0.685 mol g⁻¹ h⁻¹ and 5.475 mol g⁻¹ h⁻¹, with FE of 89% and 91% in the H-cell and the flow cell, respectively. While the presented H₂O₂ yield rate is noteworthy, the concept of employing Ni single atoms is not novel, and further material characterizations require more elaboration. Additionally, I do not see the presence of pure Ni-SAC, and the presented data actually suggests the existence of Ni nanoclusters in the matrix. In summary, considering the impressive electrochemical results, I recommend its publication in Nature Communications after a major revision. My detailed comments are as follows:

—Major comments

Comment 1. The enhancement in the intensity of the (002) peak after the acid leaching process (Fig. 1b) raises questions. It is not clear why the acid-leaching process will improve the crystalline order of CNS. Additionally, could the authors explain the observed difference in the crystalline order of the carbon matrix in Ni-SAC and Ni-CNS (red and blue curves in Fig. 1b)?

Response: We thank the reviewer for this insightful comment. We discovered that the enhancement in the intensity of the (002) peak at $\sim 25^\circ$ after acid leaching process, which may not be directly attributable to the Ni-SAC sample itself. Instead, it could be related to the glass carrier used during XRD testing. Specifically, we tested the XRD spectra of the glass carriers alone, and found that they showed significant diffraction peaks at $\sim 25^\circ$ (**Figure R1a**). Accordingly, we re-implemented the XRD tests of the Ni-SAC sample using other carriers (Frosted plastics) instead of glass to eliminate the effect on the carbon diffraction peaks. As shown in **Figure R1b**, the crystallinity of the (002) peak of Ni-SAC was significantly weakened. **The XRD pattern of Ni-SAC after re-testing have been replaced accordingly in the revised manuscript.**

Figure R1 a XRD patterns of Ni-SAC in the original manuscript and glass carrier. b XRD patterns of Ni-SAC in the revised manuscript and other carrier.

Comment 2. The HAADF-STEM image (Fig. 1e) reveals the presence of agglomerated atoms or the formation of nanoclusters. It is recommended that the

authors replace this image with a higher-quality version for clearer observation.

Response: We sincerely thank the reviewer for the professional comment. We have re-implemented HAADF-STEM to obtain higher resolution than the original manuscript image. The result in **Revised Fig. 1e** shows that the nickel exhibits uniform atomic dispersion. The corresponding image in the revised manuscript have been replaced.

Revised Fig. 1e HAADF-STEM image of Ni-SAC.

Comment 3. The elemental mapping EDX image (Supplementary Fig. 8b) depicting the distribution of Ni is not convincing and may require further clarification or improvement.

Response: Thank you for this valuable suggestion. We have provided HAADF-STEM images of Ni-SAC under different regions in a sample, which show that nickel is present in atomically dispersed state (**Revised Supplementary Fig. 8**). Meanwhile, the energy-dispersive X-ray spectroscopy (EDX) mapping results at a larger scale of Ni-SAC also reflect the dispersed distribution of nickel (**Revised Supplementary Fig. 9**).

Revised Supplementary Fig. 8 a-b HAADF-STEM image of Ni-SAC at different regions.

Revised Supplementary Fig. 9 a-e EDX mapping images for Ni-SAC.

Comment 4. The XPS data analysis requires significant revision and further clarifications. The detailed comments on the XPS are as follows:

(4-1) The authors should provide details on how they calibrated the binding energies of the XPS spectra.

Response: We thank the reviewer for this insightful comment. We are sorry that the charge correction step was missing during the processing of the XPS data in the original manuscript, which caused problems in the identification of characteristic peaks in the

original manuscript. We have re-calibrated the Ni-CNS and Ni-SAC samples with the 284.6 eV position corresponding to the adventitious carbon C (1s) spectrum as the standard (*Nat Commun.*, 2020, 11, 4246). As shown in **Figure R2**, we provide the corresponding XPS spectra of Ni-CNS as well as Ni-SAC before and after charge correction. The Ni 2p spectra of Ni-CNS and Ni-SAC after correction have positive shifts of 0.652 eV and 0.852 eV with respect to the precorrection, respectively. Accordingly, we have added the description of the XPS spectra binding energy charge correction method in the characterization section of the revised manuscript, as the following: “The binding energies were calibrated with the position 284.6 eV corresponding to the adventitious carbon.” (please see line 443, Page 24 in the revised manuscript)

Figure R2 XPS spectra of Ni 2p in **a**, Ni-CNS and **b**, Ni-SAC before and after charge correction.

(4-2) The line shapes of the peaks used to fit the XPS spectra, specifically for N 1s, S 2p, and Ni 2p, should be included in the manuscript, preferably in the Method section.

Response: We sincerely thank the reviewer for the professional comment. We provide the line shapes and the fitted curves for XPS data in the revised manuscript (including Ni 2p, N 1s, S 2p spectra; **Revised Fig.1f**, **Revised Supplementary Fig. 13** and **Revised Supplementary Fig. 14**). Fitting process of XPS data has been presented in

the Method section of the revised manuscript, as the following: “The XPS data were fitted by XPSPEAK41 and subsequently deconvoluted using Origin 2018.” (please see *Line 444, Page 24 in the revised manuscript*). The **Revised Supplementary Table 3** shown below summarizes the peak position, full width at half maximum (FWHM) width and peak area data obtained from the re-analysis of the XPS data. The corresponding Table was also provided in the revised Supplementary Information (please see *line 334, Page 25 in the Supplementary Information*).

Revised Fig.1f XPS Spectra of Ni 2p in Ni-SAC.

Revised Supplementary Fig. 13 XPS Spectra of N 1s in Ni-SAC.

Revised Supplementary Fig. 14. XPS Spectra of S $2p$ in Ni-SAC.

Revised Supplementary Table 3. Specific information on XPS data re-fitting.

Sample	XPS Peak		Position (eV)	Area	FWHM (eV)
Ni-SAC	Ni $2p$	Ni $2p_{3/2}$	853.9	2026.386	1.937
			857.0	16666.620	3.124
		Ni $2p_{1/2}$	871.0	1142.627	3.969
			874.9	7564.777	3.479
	N $1s$	pyridinic-N	398.7	361.505	3.412
		Ni-N	399.7	402.732	1.582
		pyrrolic-N	400.9	401.007	1.753
		graphitic-N	402.0	411.109	3.176
	S $2p$	Ni-S	161.5	336.754	1.940
		C-S-C $2p_{3/2}$	162.9	357.122	1.930
		C-S-C $2p_{1/2}$	164.7	449.889	2.701
		C-SO _x -C	168.6	663.133	3.063
Ni-CNS	Ni $2p$	Ni $2p_{3/2}$	853.1	1825.358	1.434
			855.8	4413.683	3.608
			870.3	834.994	1.542

		Ni 2p _{1/2}	873.2	2814.490	5.075
	N 1s	pyridinic-N	398.6	901.404	1.692
		Ni-N	399.7	758.607	2.256
		pyrrolic-N	400.9	710.530	1.222
		graphitic-N	402.0	838.416	2.430

(4-3) Assigning the peak at 853.0 eV to Ni²⁺ rather than Ni⁰ is subjected to discuss. The authors are advised to explain this assignment.

Response: Thank you for this valuable suggestion. As discussed in Comment 4-1 from Reviewer 2#, there is a problem for the XPS data during binding energy identification process due to the missing of the charge correction step in the original manuscript. After the recharge correction of XPS data, the binding energy of Ni 2p_{3/2} for Ni-CNS is located at 853.1 eV, which is attributed to zero-valent nickel (*Angew. Chem. Int. Ed.*, **2022**, 61, e202212542). The binding energy of Ni 2p_{3/2} for Ni-SAC is located at 853.9 eV, which has a 0.8 eV positive shift with respect to Ni 2p_{3/2} of Ni-CNS sample, suggesting that Ni is in an oxidation state. The corresponding discussion has been presented in the revised manuscript, as the following: “The characteristic peak in the Ni 2p_{3/2} spectra of Ni-CNS with binding energy located at 853.1 eV is assigned to zero-valent nickel (Supplementary Fig. 12)³⁶. The position of Ni 2p_{3/2} binding energy position for Ni-SAC is located at 853.9 eV, exhibiting a positive shift of 0.8 eV compared to Ni-CNS, which confirms the oxidation state of Ni in Ni-SAC (Fig. 1f).” (please see line 121, Page 6 in the revised manuscript). We also have corrected the plots relating to the Ni 2p XPS spectra (**Revised Fig. 1f, Revised Supplementary Fig.12**).

Revised supplementary Fig.12 a Ni $2p$ spectra of Ni-CNS and Ni-SAC unfitted after charge correction. **b** Ni $2p$ XPS spectra of Ni-SAC, Ni-CNS.

(4-4) The inconsistency in the peak ratio under Ni $2p_{3/2}$ relative to what is fitted under Ni $2p_{1/2}$ needs clarification. Concerning the quantum states, i.e., $2j+1$ for $j=1/2$ and $3/2$, the relative intensity ratio of Ni $2p_{3/2}$ to Ni $2p_{1/2}$ should be 2 for each oxidation state.

Response: We thank the reviewer for this insightful comment. We are sorry that there were irregularities in curve-fitting of the original manuscript (*please see Comment 4-1 and 4-3 from Reviewer 2#*), so we have re-implemented the XPS fitting analysis. After more rigorous analytical processing of the XPS data, the relative intensity ratio of Ni $2p_{3/2}$ to Ni $2p_{1/2}$ is close to 2:1. Taking Ni $2p_{3/2}$ and Ni $2p_{1/2}$ of Ni-SAC as the example, the peak area ratios are $2026.386:1142.627=1.8:1$ and $16666.62:7564.777=2.2:1$, respectively, which are close to the ratio of 2:1. Specific fitting information we have given in **Revised Supplementary Table 3**, where the Ni $2p$ spectra were summarized again for reviewers in **Table R1**.

Table R1 Specific information on XPS data fitting for Ni $2p$.

Sample	XPS Peak		Position (eV)	Area	FWHM (eV)
Ni-SAC	Ni $2p$	Ni $2p_{3/2}$	853.9	2026.386	1.937
			857.0	16666.620	3.124

		Ni 2p _{1/2}	871.0	1142.627	3.969
			874.9	7564.777	3.479
Ni-CNS	Ni 2p	Ni 2p _{3/2}	853.1	1825.358	1.434
			855.8	4413.683	3.608
		Ni 2p _{1/2}	870.3	834.994	1.542
			873.2	2814.490	5.075

(4-5) I suggest that the authors re-conduct the XPS analysis for both Ni-SAC and Ni-CNS samples with a wider range to obtain a more accurate baseline. Additionally, they should re-fit the data and provide information about the line shapes in the Supplementary Information.

Response: We sincerely thank the reviewer for the professional comment. In response to the reviewers' comments, we re-baselined all XPS data over a wider range of intervals and re-fitted the curves. A larger range of baseline gives helps to improve the fit of the curve and give more accurate analysis results. The re-fitted analyzed data are shown in **Figure R3**, and we have updated the corresponding plots to **Revised Fig. 1f** for the revised manuscript and **Revised Supplementary Fig. 12-15** for the Supplementary information. The specific information on the re-fitted XPS data is shown in **Revised Supplementary Table 3** (*please see the response to Comment 4-2 from Reviewer 2#*).

Figure R3 XPS spectra of **a**, Ni 2*p* **b**, N 1*s* and **c**, S 2*p* in Ni-SAC, respectively. XPS spectra of **d**, Ni 2*p* and **e**, N 1*s* in Ni-CNS.

Comment 5. The presence of a shoulder at 2.2 Å (Ni-Ni) in the Ni-SAC sample suggests the existence of Ni nanoclusters (Fig. 1h). XAS data does not indicate the presence of pure Ni single atoms in the matrix. I additionally recommend that the authors provide a model to fit the Fourier-transformed EXAFS spectra.

Response: Thank you for this valuable suggestion. We performed wavelet transform (WT) to confirm whether the Ni-Ni bond exists in Ni-SAC, as the following: “The Wavelet transform EXAFS (WT-EXAFS) show that the maximum intensity of the WT contour plot for Ni-SAC occurs around 4.35 Å⁻¹, which is significantly distinct from that of Ni foil (7.5 Å⁻¹) and NiO (6.8 Å⁻¹), confirming the atomic dispersion of Ni elements in Ni-SAC (Supplementary Fig. 15). Moreover, the WT contour maximum intensity of Ni-CNS is located at 7.3 Å⁻¹, close to that of Ni foil, echoing the presence of zero-valent Ni as indicated by its XPS analysis.” (please see line 140, Page 7 in the revised manuscript)

Revised Supplementary Fig. 15 WT-EXAFS spectra of **a**, Ni foil, **b**, NiO, **c**, Ni-SAC and **d**, Ni-CNS.

In view of you proposed the shoulder at 2.2 Å, since the presence of the Ni-Ni bond has been ruled out, we supposed that it may be due to the coordination of the Ni single atom with different heteroatoms. It is noteworthy that the main peak of Ni-SAC is located at the higher R -space of 1.75 Å unlike the widely reported Ni-N₄ (1.49 Å) (**original Fig. 1h**) (*Angew. Chem. Int. Ed.*, **2023**, 62, e202216511). Considering the successful introduction of N and S into the coordination environment as shown by XPS analysis (**Revised Supplementary Fig. 13** and **Revised Supplementary Fig. 14**), we speculate that Ni-SAC is the structure in which Ni is co-coordinated with light heteroatoms (N/S). The corresponding discussion has been presented in the revised manuscript (*please see line 135, Page 7 in the revised manuscript*). The optimal fitting results of quantitative least-squares EXAFS curve-fitting analysis show that the Ni atom is coordinated with 3.9 heteroatoms (N/S) in the first shell (**Revised Supplementary Fig. 16** and **Revised Supplementary Table 4**). The corresponding discussion has been presented in the revised manuscript (*please see line 145, Page 7 in the revised manuscript*).

Revised Supplementary Fig. 16 The k^2 -weighted Fourier transformations (FT)-EXAFS spectra and FT-EXAFS fitting plots of **a**, Ni-CNS and **b**, Ni-SAC.

Revised Supplementary Table 4 Curve-fitting results of Ni K-edge EXAFS spectra.

Sample	Shell	R (Å)	CN	$\sigma^2(10^{-3}\text{Å}^2)$	$\Delta E^0(\text{eV})$	R factor (%)
Ni-SAC	Ni-N/S	2.06	3.9	3.5	8.5	0.9
Ni-CNS	Ni-N/S	2.01	4.9	1.6	8.3	1.1
	Ni-Ni	2.48	5.6	6.4		
Ni-foil	Ni-Ni	2.48	11.5	6.0	6.1	0.1
NiO	Ni-O	2.07	5.4	7.5	3.1	0.6
	Ni-Ni	2.95	11.0	7.5		

CN is the coordination number; R is the distance between absorber and backscatter atoms; σ^2 is Debye-Waller factor value; ΔE_0 is inner potential correction accounts for the difference in the inner potential between the sample and the reference compound; R -factor indicates the goodness of the fit. Fitting range: $2.5 \leq k$ (Å^{-1}) ≤ 11.0 and $1.0 \leq R$ (Å) ≤ 3 .

Comment 6. Since XRD cannot show the presence of nanoclusters, I recommend

presenting STEM images after the stability test to examine the presence of any agglomerated Ni clusters.

Response: We thank the reviewer for this insightful comment. We supplemented the HAADF-STEM image of Ni-SAC sample after reaction. As shown in the **Revised Supplementary Fig. 23**, Ni-SAC can maintain the atomically dispersed state of Ni after the stability test (operate stably for 20 h at 0.3 V vs. RHE). The corresponding discussion has been presented in the revised manuscript. (*please see line 188, Page 10 in the revised manuscript*).

Revised Supplementary Fig. 23 HAADF-STEM images of Ni-SAC after reaction.

Comment 7. The observed improvement in the electrochemical activity of Ni-SAC compared to Ni-CNS could be attributed to the higher surface area of Ni-SAC, as indicated by SEM images and BET data. Therefore, I suggest that the authors measure the electrochemical surface area of these samples and include a normalizing of the current based on this factor in the Supplementary Information.

Response: We sincerely thank the reviewer for the professional comment. We carry out electrochemical active surface area (ECSA) experiment to reveal the intrinsic activity of Ni-SAC. The CV tests were implemented at the scan rates of 20, 40, 60, 80, and 100 mV s^{-1} in the non-Faradaic interval in 0.1 M N_2 -saturated KOH to exclude the effect of trace oxygen in the electrolyte. Plotting Δj as a function of scan rate, the slope of which is the double layer capacitance (C_{dl}). (*Nat. Commun.*, **2024**, 15, 983, *J. Am. Chem.*

Soc., **2013**, 135, 45, 16977). The corresponding discussion has been presented in the method of revised manuscript. (please see line 512, Page 27 in the revised manuscript) As shown in **Supplementary Fig. 19a-c**, Ni-SAC has a larger C_{dl} relative to Ni-CNS (33.5 mF cm^{-2} vs. 21.8 mF cm^{-2}), suggesting that the former possesses a larger ECSA thereby enhancing the utilization of active sites. We further employed ECSA normalized LSV curves as suggested by the reviewer to rule out the contribution of large surface area to the activity for Ni-SAC (**Supplementary Fig. 19d**). Compared to Ni-CNS, Ni-SAC maintains higher current density over the measured potential interval proving its higher intrinsic activity. The corresponding discussion has been presented in the revised manuscript. (please see line 172, Page 9 in the revised manuscript and line 61, Page 9 in the revised Supplementary Information)

Revised Supplementary Fig. 19 Cyclic voltammetry (CV) of **a**, Ni-SAC and **b**, Ni-CNS in non-Faradaic interval at the scan rates of 20, 40, 60, 80 100 mV s^{-1} . **c** Curves of capacitance Δj as a function of different scan rates. **d** LSV curves normalized by ECSA.

$$\text{ECSA} = \frac{C_{dl}}{C_s}$$

C_{dl} is the measured double-layer capacitance, C_s is the specific capacitance of an atomically smooth planar surface, in which the general value of C_s is 0.040 mF cm^{-2}

(*Nat. Commun.*, **2023**, 14, 368).

Comment 8. I recommend that the authors calculate and include the number of consumed electrons per active site per second versus E, based on the method proposed in Nat Commun 13, 1256 (2022), for both H-cell and the flow cell in the Supplementary Information, with a corresponding note in the main manuscript.

Response: We thank the reviewer for this insightful comment. According to the reviewers' comments, we provide the variation of TOF values (The formulae related to the TOF values has been illustrated in **Supplementary Note 1.**) and the corresponding consumed electron rate (R_e) versus E in both H-cell and flow cell devices.

R_e is defined as the total number of electrons consumed per second per the number active sites (*Nat. Commun.*, **2022**, 13, 1256), which for the $2e^-$ ORR reaction is defined as:

$$R_e = 2 * TOF$$

As shown in **Revised Supplementary Fig. 30**, the R_e value increases gradually with the overpotential in both H-cell and flow cell, which has a more rapid enhancement in the flow cell than that in the H-cell. The maximum R_e value that can be achieved in the flow cell is approximately 7.56 times that of the H-cell. The corresponding discussion has been presented in the revised manuscript as follows: “Furthermore, we calculated the electron consumption rate (R_e) of the constructed system, and the maximum R_e value that can be achieved in the flow cell is about 7.56 times higher than that in the H-cell ($3.76 \text{ e}^- \text{ s}^{-1}$ vs. $28.52 \text{ e}^- \text{ s}^{-1}$).” (please see line 254, Page 13 in the revised manuscript)

Revised Supplementary Fig. 30 TOF values and R_e of Ni-SAC at different voltages in **a**, H-cell and **b**, flow cell, respectively.

—Minor comments

Comment 1. CNS should be defined in both the Abstract (Line 18) and Introduction (Line 72) sections.

Response: We thank the reviewer for this insightful comment. We have added the definition of CNS (carbon nanosheets) in the reworked manuscript. (*please see Line 19, page 1 and Line 73, page 4 in the revised manuscript*)

Comment 2. The sentence 'Pure carbon nanosheet ... condition for 12 h (Supplementary Fig. 3)' lacks clarity and needs to be rewritten for better understanding.

Response: Thank you for this valuable suggestion. We have rewritten the sentence in the revised manuscript for better comprehension, as the following: “Carbon nanosheets (CNS) electrode almost without Ni was also obtained by acid etching the Ni-CNS in 1 M HCl under heating condition (60 °C) for 12 h (Supplementary Fig. 3 and Supplementary Table 2).” (*please see Line 89, page 5 in the revised manuscript*).

Comment 3. Orbital symbols should be written in italic font.

Response: We have re-examined the standards of orbital symbol writing and corrected them. (For example, *Line 121 Page 6*: Ni $2p_{3/2}$, *Line 125 Page 6*: N $1s$)

Comment 4. The authors are advised to plot the raw data of the XPS spectra (Fig. 1f, Supplementary Fig. 11, Supplementary Fig. 12, and Supplementary Fig. 13) with a thicker line to facilitate the examination of noise and fluctuations.

Response: We thank the reviewer for this insightful comment. We have bolded the raw XPS data in order to better check the fitting peaks. As shown in **Figure R4**, we provide two sets of baseline bolding examples for reviewers' reference.

Figure R4 Ni $2p$ XPS spectra after baseline bolding.

Comment 5. The N $1s$ spectrum of Ni-CNS is missing. The authors should include the N $1s$ spectrum of Ni-CNS in the Supplementary Information.

Response: We sincerely thank the reviewer for the professional comment. As shown in the **Revised Supplementary Fig. 13**, the XPS spectrum of N $1s$ in Ni-CNS can be deconvoluted into four peaks: pyridine-N (398.6 eV), Ni-N (399.7 eV), pyrrolic-N (400.9 eV), and graphite N (402.0 eV). (*Angew. Chem. Int. Ed.* **2022**, 61, e202203335, *Angew. Chem. Int. Ed.* **2020**, 59, 13057).

Revised Supplementary Fig. 13 XPS spectra of N 1s in Ni-CNS.

Comment 6. Considering the variation in loaded mass across different studies, I recommend the authors plot H_2O_2 yield ($\text{mmol h}^{-1} \text{cm}^{-2}$) versus E , incorporating data from other studies in the Supplementary Information, similar to Fig. 2d.

Response: We thank the reviewer for this insightful comment. We give a comparison of the corresponding H_2O_2 yields ($\text{mmol cm}^{-2} \text{h}^{-1}$) versus E in the currently reported work under alkaline condition (0.1 M KOH) in H-cell. After accounting for the different loadings of the catalyst, Ni-SAC still exhibits impressive H_2O_2 yield rate relative to most of the work reported to date (**Revised Supplementary Fig. 21 and Revised Supplementary Table 5**). The corresponding Figure and Table have been added to the revised Supplementary Information.

Revised Supplementary Fig. 21 Comparison of H₂O₂ yield rate (mmol cm⁻² h⁻¹) with previous reported works under alkaline condition (0.1 M KOH) in H-cell.

Original Fig. 2d Comparison of H₂O₂ yield rate (mmol g_{cat}⁻¹h⁻¹) with other reported literature under alkaline condition (0.1 M KOH) in H-cell.

Revised Supplementary Table 5 Comparison of 2e⁻ORR performance in H-cell under alkaline conditions (0.1 M KOH) with previous reported works.

No.	Catalyst	Yield rate (mmol cm ⁻² h ⁻¹)	Yield rate (mmol g _{cat} ⁻¹ h ⁻¹)	Ref.
1	Co-SAs/NC	/	38.1 (@0.5 V)	Inorg. Chem. Front. , 2021 , 8, 2829
2	Mo-TiO ₂ -2	~0.025 (@0.2 V)	~250 (@0.2 V)	Catal. Sci. Technol. , 2021 , 11, 6970
3	Oxo-G/NH ₃ ·H ₂ O	0.02248 (@0.2 V)	224.8 (@0.2 V)	ACS Catal. , 2019 , 9, 1283
4	Ni ₄ -B ₁ @BNC	0.0655 (@0.6 V)	128.5 (@0.6 V)	Small , 2022 , 18, 22035
5	PEI50CMK3_800T	0.0173 (@0.2 V)	345.5 (@0.2 V)	ChemSusChem. , 2018 , 11, 3388
6	Ni MOF NSs-6	0.09 (@0.3 V)	90 (@0.3 V)	Angew. Chem. Int. Ed. , 2021 , 60, 11190

7	O-BC-2-650	~ 0.083 (@0.55 V)	412.8 (@0.55 V)	Sci. China Mater. , 2022 , 65, 1276
8	Ni-N ₂ O ₂ /C	~ 0.0104 (@0.4 V)	45.1 (@0.4 V)	Angew. Chem. Int. Ed. , 2020 , 59, 13057
9	a-NiO NSs	0.03625 (@0.51 V)	145 (@0.51 V)	Cell Rep. Phys. Sci. , 2022 , 3, 100788
10	N-DCDs	/	613.58 (@0.3 V)	J. Mater. Chem. A. , 2023 , 11, 11704
11	MgP-DHTA-COF	0.0724 (@0.2 V)	362 (@0.2 V)	Chem. Asian J. , 2021 , 16, 498
12	Ni-SAC	0.2904 (@0.2 V)	726.1 (@0.2 V)	This work

Comment 7. Could the authors provide the error associated with the desorption peak in the O₂-TPD results (Fig. 4e)?

Response: We thank the reviewer for this insightful comment. According to the suggestion, we give the error analysis of the O₂-TPD results after three parallel experiments. As shown in **Revised Fig. 4e**, the highest desorption temperature (146.5 °C) and oxygen desorption amount of Ni-CNS show that the sample has the highest adsorption degree of O₂. The desorption temperatures of Ni-SAC and CNS are approach (139.4 °C vs. 136.5 °C), while the corresponding oxygen desorption quantities of the two samples are weakened in turn, which indicated that the adsorption capacity of Ni-SAC for O₂ is moderate. The corresponding Figure and discussion have been presented in the revised manuscript. (*please see Line 312, page 16 in the revised manuscript*)

Revised Fig. 4e O₂-TPD for the Ni-SAC, Ni-CNS, CNS.

The results of three parallel experiments for each sample are shown in **Figure R5**.

Figure R5 a-i O₂-TPD triple test results for three samples.

REVIEWER COMMENTS

Reviewer #1 (Remarks to the Author):

The authors have properly addressed the reviewer's comments and the manuscript is recommended for publication.

Reviewer #2 (Remarks to the Author):

The authors have done an excellent job in addressing most of my comments. However, in the Method section, they need to clarify which lineshape function (such as Gaussian, Lorentzian, Voigt, etc.) was used to fit the XPS data. Additionally, there is a minor issue with the baselines of the N 1s and S 2p (as shown in Supplementary Fig. S13a and Fig. S14) that needs to be corrected. In my opinion, the current baseline is incorrect. Once these issues are addressed, the peak information in the Supplementary Table S1 to S3 will need to be modified accordingly. Overall, I believe that if these changes are made, the manuscript will be suitable for publication in Nature Communications.

We appreciate the reviewers for their valuable comments. Details of the corresponding revisions are described below point by point.

Reviewer 1#

The authors have properly addressed the reviewer's comments and the manuscript is recommended for publication.

Response: Thank the reviewer for your recognition of our work.

Reviewer 2#

The authors have done an excellent job in addressing most of my comments. However, in the Method section, they need to clarify which lineshape function (such as Gaussian, Lorentzian, Voigt, etc.) was used to fit the XPS data. Additionally, there is a minor issue with the baselines of the N 1s and S 2p (as shown in Supplementary Fig. S13a and Fig. S14) that needs to be corrected. In my opinion, the current baseline is incorrect. Once these issues are addressed, the peak information in the Supplementary Table S1 to S3 will need to be modified accordingly. Overall, I believe that if these changes are made, the manuscript will be suitable for publication in Nature Communications.

Response: We sincerely thank the reviewer for the professional comment.

- (1) The XPS data is fitted by using the mixed Gaussian-Lorentzian function, as reported in the previous reports (*ACS Catal.*, **2023**, 13, 15301, *Appl. Surf. Sci.*, **2018**, 447, 548). The corresponding discussion has been presented in the revised manuscript, as the following: “The fitting of spectral peaks in the XPS data was performed using the XPSPEAK41 software, which applied a mixed Gaussian-Lorentzian function.”
(please see line 443, Page 24 in the Method section of revised manuscript)
- (2) For the baseline of N 1s and S 2p spectra for Ni-SAC, we reconfirmed the selection of baseline to obtain a more accurate XPS fit analysis (**Revised Supplementary**

Fig. 13a and Fig. 14). The basic principle of baseline confirmation is to pass through the middle of the data noise before and after the peaks (*J. Vac. Sci. Technol. A.*, 2020, 38, 061203). The results of the reprocessing of the XPS data are shown below, and the replacement of the corresponding figures and table in the supplementary information has been completed.

Revised Supplementary Fig. 13a XPS Spectra of N 1s in Ni-SAC.

Revised Supplementary Fig. 14. XPS Spectra of S 2p in Ni-SAC.

Revised Supplementary Table 3. Specific information on XPS data re-fitting.

Sample	XPS Peak		Position (eV)	Area	FWHM (eV)
Ni-SAC	Ni 2p	Ni 2p _{3/2}	853.9	2026.386	1.937
			857.0	16666.62	3.124
		Ni 2p _{1/2}	871.0	1142.627	3.969
			874.9	7564.777	3.479
	N 1s	pyridinic-N	398.5	137.783	1.489
		Ni-N	399.8	408.814	1.451
		pyrrolic-N	400.8	261.671	1.459
		graphitic-N	401.8	228.294	1.951
	S 2p	Ni-S	161.7	240.370	1.817
		C-S-C 2p _{3/2}	163.0	407.669	2.091
		C-S-C 2p _{1/2}	164.8	245.341	2.508
		C-SO _x -C	168.6	405.683	3.398
Ni-CNS	Ni 2p	Ni 2p _{3/2}	853.1	1825.358	1.434
			855.8	4413.683	3.608
		Ni 2p _{1/2}	870.3	834.994	1.542
			873.2	2814.490	5.075
	N 1s	pyridinic-N	398.5	652.840	1.508
		Ni-N	399.7	410.720	1.584
		pyrrolic-N	400.8	683.506	1.164
		graphitic-N	401.7	750.613	2.417

REVIEWERS' COMMENTS

Reviewer #2 (Remarks to the Author):

The authors have clarified everything and their paper can be suggested for publication in Nature Communications.